# The Ice Cloud Imager: retrieval of frozen water column properties

Eleanor May[1], Bengt Rydberg[2], Inderpreet Kaur[1,2], Vinia Mattioli[3], Hanna Hallborn[1], and
Patrick Eriksson[1]

[1]Department of Space, Earth and Environment, Chalmers University of Technology, Gothenburg, Sweden
[2]Swedish Meteorological and Hydrological Institute (SMHI), Norrköping, Sweden
[3]EUMETSAT, Darmstadt, Germany

**Correspondence:** Eleanor May (eleanor.may@chalmers.se)

**Abstract.**

The Ice Cloud Imager (ICI) aboard the Second Generation of the EUMETSAT Polar System (EPS-SG) will provide novel measurements of ice hydrometeors. ICI is a passive conically scanning radiometer that will operate within a frequency range of 183 GHz to 664 GHz, helping to cover the present wavelength gap between microwave and infrared observations. Reliable global data will be produced on a daily basis. This paper presents the retrieval database to be used operationally and performs a final pre-launch assessment of ICI retrievals.

Simulations are performed within atmospheric states that are consistent with radar reflectivities and represent the three-dimensional variability of clouds. The radiative transfer calculations use empirically-based hydrometeor models. Azimuthal orientation of particles is mimicked, allowing for polarisation to be considered. The degrees of freedom of the ICI retrieval database are shown to vary according to cloud type. The simulations are considered to be the most detailed performed to this date. Simulated radiances are shown to be statistically consistent with real observations.

Machine learning is applied to perform inversions of the simulated ICI observations. The method used allows for the estimation of non-Gaussian uncertainties for each retrieved case. Retrievals of ice water path (IWP), mean mass height ($Z_{\mathrm{m}}$) and mean mass diameter ($D_{\mathrm{m}}$) are presented. Distributions and zonal means of both database and retrieved IWP show agreement with DARDAR. Retrieval tests indicate that ICI will be sensitive to IWP between $10^{-2}$ and $10^{1}$ kg m$^{-2}$. Retrieval performance is shown to vary with climatic region and surface type, with the best performance achieved over tropical regions and over ocean. As a consequence of this study, retrievals on real observations will be possible from day one of the ICI operational phase.

## 1 Introduction

There is an urgent need for reliable and consistent estimates of ice mass in the atmosphere. Measurements of ice mass are pivotal to understanding global weather patterns and current climate trends. The release of latent heat upon ice formation drives circulation within the atmospheric water cycle (Bony et al., 2015), giving rise to events such as deep convective systems. Frozen hydrometeors also constitute the ice clouds in our atmosphere. Such clouds modulate the amount of outgoing longwave radiation (OLR) and reflect solar radiation, and thus hold significant influence over the climate. Consequently, ice clouds are considered a major climate feedback (Stephens et al., 1990).

Despite their importance, the mass of ice hydrometeors in the atmosphere remains an uncertain quantity. A significant factor contributing to this uncertainty is the complex relationship between ice mass and satellite observation — the relationship is difficult to characterise, thus leading to a large spread between models and data-derived observational products, and even between the products themselves. Duncan and Eriksson (2018) highlighted these shortcomings by showing the significant discrepancies that exist between cloud ice datasets. Consequently, ice hydrometeor mass observations remain a gap in the atmospheric hydrological cycle. Given the present-day state of the climate, it is becoming increasingly necessary to address this gap (Waliser et al., 2009).

The upcoming Ice Cloud Imager (ICI) will address this issue by specifically measuring atmospheric ice mass at microwave and sub-millimetre wavelengths (Mattioli et al., 2019). ICI will be launched as part of the EUMETSAT (The European Organisation for the Exploitation of Meteorological Satellites) Polar System - Second Generation (EPS-SG) programme. While current satellite observations span a wide range of frequencies, many face limitations in the measurement of ice hydrometeors. High-frequency radar has good sensitivity to ice, but are limited to observe at only a single frequency and thus poorly constrain hydrometeor properties. Passive optical and infrared missions, such as MODIS (Moderate-resolution Imaging Spectroradiometer; Platnick et al. 2003) and SEVIRI (Spinning Enhanced Visible and Infrared Imager; Schmid, 2000) are sensitive to small ice hydrometeors. However, these missions predominantly capture only cloud top data due to high attenuation. Passive microwave sensors are able to penetrate cloud, but measure at wavelengths that are sensitive only to the largest ice hydrometeors. Sub-millimetre frequencies are most suited to the measurement of small ice crystals. However, sub-millimetre observations of the atmosphere are scarce.

ICI will pair sub-millimetre wavelength observations with passive microwave observations. This allows for sensitivity to a wider range of ice hydrometeor sizes, and increases the overall information content through the use of multiple frequencies. ICI will cover a frequency range of 183 GHZ to 664 GHz. Alongside ICI on the same satellite will be the MWI (The Microwave Imager), extending the coverage down to 18.7 GHz. Together, ICI and MWI will offer an unparalleled collection of passive measurements, helping to close the existing frequency gap in ice hydrometeor mass observations (Accadia et al., 2020).

Evidence supports the improvement of cloud ice retrievals in the presence of sub-millimetre observations. Successful cloud ice retrievals have been performed using sub-millimetre limb sounder observations (Wu et al., 2008; Eriksson et al., 2014). Brath et al. (2018) demonstrated that retrievals of ice water path (IWP) improved when combining microwave and sub-millimetre flight campaign observations. Pfreundschuh et al. (2022b) showed enhanced sensitivity to hydrometeors when supplementing radar data with sub-millimetre observations. In the context of a future ICI launch, Wang et al. (2016) performed successful retrievals of cloud parameters based on a synthetic database of ICI radiances. Finally, Eriksson et al. (2020) — the precursor to this study — presented the operational algorithm to be used within the ICI Level 2 (L2) product at EUMETSAT. In the study, a preliminary database of simulated ICI observations was used to demonstrate successful ICI retrievals and to estimate the retrieval performance.

This study presents the generation of the final retrieval database, composed of ICI observation simulations and corresponding ice mass quantities. This database serves to complete the operational algorithm, and performs a pre-launch characterisation of ICI retrievals. We emulate retrieval performance using synthetic ICI data, where retrievals meet operational standards, i.e. only

ice hydrometeors are considered. The study focuses only on channels present on ICI, and MWI measurement frequencies are not considered. The primary aim of the study is to confirm that both simulations and retrievals are statistically consistent with reality. In the absence of actual 'ground truth', flight-campaign data and existing retrieval products are used. Additionally, we investigate how retrieval performance varies according to climatic region, surface conditions, and noise estimates. In summary, we perform the best possible pre-launch assessment of the retrieval of ice hydrometeor column values. To this end, we evaluate whether the ICI L2 product is set to deliver reliable information on ice mass in the form of ice water path (IWP), mean mass height ($Z_m$) and mean mass diameter ($D_m$), and under which conditions.

A description of the retrieval database and the database generation techniques are given in Sect. 3. Details on the retrieval approach are given in Sect. 4. In Sect. 5, radiances resulting from simulations of two similar instruments are compared with real observations, and an assessment of the overall retrieval performance is given.

## 2 The Ice Cloud Imager (ICI) Mission

### 2.1 The ICI instrument

ICI will be hosted on the MetOp-SG B satellite series as part of the EPS-SG mission. The mission will launch a pair of satellites: MetOp-SG A and B. MetOp-SG A will primarily focus on visible and infrared observations, while MetOp-SG B will host microwave instruments (EUMETSAT, 2022). Each satellite has a planned lifetime of 7.5 years. With three successive launches of the pair of satellites, continuous coverage for over 21 years will be achieved. The satellites will be in a sun-synchronous orbit with local time of descending node at 9:30, and will fly at a height within the range of 823 - 843 km. At the time of writing, the launch of the first satellite in each of the two EPS-SG series is scheduled for 2025 and 2026, respectively.

The ICI instrument is a conically scanning radiometer, providing observations at an incidence angle of $53° \pm 2°$. Observations are taken over an angle of $\pm 65°$ around the forward view of the orbital path, giving a swath width of approximately 1700 km, which results in near-global coverage on a daily basis. ICI will observe using 13 channels, operating at local oscillator frequencies of 183.31, 243.2, 325.15, 448.0, and 664.0 GHz. Specifications of the channels can be found in Eriksson et al. (2020). Nine channels around 183.31, 325.15 and 448.0 GHz cover water vapour molecule transitions. The remaining channels at 243.2 and 664.0 GHz function as 'window' channels, where decreased absorption by atmospheric gases enables observations down to relatively low altitudes. The window channels will measure at both horizontal and vertical polarisation. As such, ICI has the potential to capture the effects of oriented hydrometeors and of polarisation as a result of surface interaction. The other channels (183.31, 325.15 and 448.0 GHz) will measure at only vertical polarisation. Further details on the receivers, the antenna system and calibration are given in Eriksson et al. (2020). However, some updates have been made since the publication of Eriksson et al. (2020); detailed estimates of the spectral response function and the antenna response function are now available. Furthermore, first actual estimates of the noise performance are now at hand, indicating that the noise requirements will be met with a margin. Consequently, we assume below that the magnitude of thermal noise is at 75% of the NE$\Delta$T estimates given in Eriksson et al. (2020).

Also on board MetOp-SG B will be MWI — another conically scanning radiometer that will also observe at an incidence angle of $\approx 53°$. MWI will measure at frequencies between 18.7 GHz and 183 GHz, allowing for sensitivity to liquid and frozen precipitation. It will offer both horizontal and vertical polarisation up to frequencies of 89 GHz, and vertical polarisation only for the higher-frequency channels.

## 2.2 Operational L2 product

EUMETSAT will offer a Level 2 product (MWI-ICI-L2) containing retrievals based on ICI and MWI observations in near real-time. Although retrievals will be performed separately for ICI and MWI using different methods, they will be present in the same product. This is the only operational L2 product planned. As a result of the MWI contribution, MWI-ICI L2 will offer liquid water path (LWP). The primary variable offered on the ICI-side of the MWI-ICI L2 product is ice water path (IWP), defined as the vertical column of atmospheric ice water content. Mean mass height ($Z_\mathrm{m}$) and mean mass diameter ($D_\mathrm{m}$) are also offered as retrieval products for ICI, existing only for IWP $> 0 \ \mathrm{kg \ m^{-2}}$. This study deals only with the ICI-side variables, of which definitions are given in Eriksson et al. (2020). The algorithm theoretical basis document (ATBD) is provided by the EUMETSAT Satellite Application Facility (SAF) supporting nowcasting (NWC-SAF) (Rydberg, 2018). The algorithm is finalised. What remains for the L2 product is the development of a database (discussed below), which is of major importance for retrieval accuracy.

Within the algorithm, pre-processing is undertaken before the inversion takes place. The pre-processing steps include the remapping of Level 1b (L1b) data, i.e. calibrated and geo-located antenna temperatures, to a common footprint, that was the output of an optimal interpolation scheme developed within a EUMETSAT study with results presented in Eriksson et al. (2020). Additionally, there is a bias correction scheme applied in the case of any systematic differences between L1b data and retrieval database cases. An optional module for the detection of clear-sky cases, i.e. IWP = 0, is also included within the algorithm. The reader is referred to Eriksson et al. (2020) for further details of the pre-processing steps.

To perform the inversion, the retrieval algorithm uses an implementation of Bayesian Monte Carlo integration (BMCI) (Eriksson et al., 2020). The inversion itself is centred around a retrieval database — a dataset containing ice mass products and associated observations. The aim of the algorithm is to map observations to ice mass products, using the retrieval database as reference data. The input to the algorithm is a vector of measurements, and the outputs are the retrieval products (IWP, $Z_\mathrm{m}$, $D_\mathrm{m}$) associated with the given measurements. The measurements take the form of geolocated and calibrated antenna temperatures. Alternatively, the measurement vector can be replaced with a cloud signal, i.e. a simulated clear-sky observation subtracted from the original all-sky case. The ICI retrieval algorithm takes the latter approach. This has the advantage of reducing the background contribution to the observation, where Rydberg (2018) can be referred to for more detail.

In Rydberg (2018), a retrieval database was developed with the aim of testing the ICI retrieval algorithm and estimating retrieval performance. Although the simulations were detailed, this preliminary database was limited in some aspects. Particle orientation effects and three-dimensional variability within clouds were not accounted for. The purpose of this study is the development of a new and final retrieval database for operational use in the L2 product and an updated characterisation of expected retrievals. We refer to the database of this previous study as the *preliminary database*. The database generation

method has been updated to reflect any recent developments, and the limitations of the preliminary database are addressed. This is discussed further in the following section. The database generated in this study is referred to as the *new database* or, simply, the retrieval database. The development of the operational L2 product at EUMETSAT was finalised with the completion of this new retrieval database.

## 2.3  Numerical weather prediction

The main application of ICI is to provide global measurements related to ice clouds in support of climate monitoring. Another application area of ICI is numerical weather prediction (NWP). ICI could be of some interest for assimilation systems of clear-sky character; its radiances would provide some additional information on humidity, or could be used for cloud filtering (Kaur et al., 2021) of MWI data. However, as ICI is designed to constrain ice hydrometeor properties (Buehler et al., 2007), its data will be most useful if used in all-sky assimilation. As a consequence, the preparatory work inside NWP and developing L2 products has considerable overlap, both applications require development of radiative transfer involving scattering of ice particles.

A common step in this direction was the EUMETSAT study that resulted in the single-scattering data presented in Eriksson et al. (2018), which was developed for use within The Atmospheric Radiative Transfer Simulator (ARTS) software (Buehler et al., 2018). This "ARTS database" is not only used for the L2 retrievals of concern here, but parts of it are now also incorporated in the leading NWP forward operators RTTOV (Radiative Transfer for TOVS) (Geer et al., 2021) and the Community Radiative Transfer Model (CRTM) (Moradi et al., 2022). One aim of Eriksson et al. (2018) was to offer a broad set of particle shapes, and for practical issues effects of particle orientation could not be accommodated in this relatively large database. As a complement, Brath et al. (2020) covers particle orientation for two habits. The later data were developed for supporting research and L2 retrievals, but turned out to be vital for developing a fast scheme to approximate the impact of particle orientation (Barlakas et al., 2021) inside RTTOV-SCATT (the microwave all-sky module of RTTOV). This development, in its turn, triggered an extension of the fast scheme that became used in the production of the new retrieval database (Sect. 3).

The ARTS software itself is supporting NWP by acting as a reference model (Barlakas et al., 2022a). Reversely, the new microwave absorption model developed for RTTOV is applied when running ARTS inside this study (Sect. 3.1). Largely based on these efforts, there is already infrastructure in place at the European Centre for Medium-Range Weather Forecasts (ECMWF) to make use of ICI radiances in all-sky assimilation (Alan Geer, ECMWF, private communication).

## 3  Retrieval database

The novelty of ICI lies in its ability to provide new and unique observations at sub-millimetre wavelengths. However, this presents a challenge when creating a retrieval database, given the current unavailability of sub-millimetre observations. The retrieval database must be assembled in the absence of actual observations, preventing the creation of an empirically-based database. Instead, it is necessary to simulate ICI observations.

There have been several attempts at generating retrieval databases of simulated sub-millimetre observations. One early use of a retrieval database is Evans et al. (2002). Stochastic cloud profiles were constructed based on the in-situ microphysical data, which are then used within radiative transfer calculations. The result was simulated brightness temperatures as they would be measured by the aircraft-mounted Submillimetre-Wave Cloud Ice Radiometer (SWCIR). A later attempt to develop a retrieval database is detailed in Rydberg et al. (2007). This database was limited to one-dimensional atmospheric states, used only single frequency channels, and was limited to northern mid-latitudes. The method was later extended (Rydberg et al., 2009) to include CloudSat (Stephens et al., 2002) data in the generation of three-dimensional atmospheric scenes using a Fourier transform algorithm, but simplified cloud microphysics remained a limitation. Evans et al. (2012) performed IWC retrievals using sub-millimetre flight campaign data. The retrieval database was built upon stochastic hydrometeor profiles that were generated using CloudSat radar reflectivities and microphysics probability distributions. The microphysical information was based on in-situ data, and is therefore detailed but limited by the region of flight and the location of in-cloud sampling. In Wang et al. (2016), a synthetic database of ICI observations was created by performing radiative transfer calculations on simulated hydrometeor profiles. The database was successfully used to perform cloud ice retrievals specifically for ICI, but was limited to cases over Europe. Finally, the preliminary database developed by Eriksson et al. (2020) marked the most recent major development. The database reflects global variability with regard to atmospheric and surface scenarios. Additionally, the microphysical assumptions were improved. However, some areas remained to be improved upon for the new retrieval database.

## 3.1 Overview

The reliability of the retrievals relies strongly on the quality of the retrieval database. As such, there are several requirements placed on the database. Firstly, the database must contain realistic simulations of the future ICI observations, thus considering all relevant atmospheric, surface and instrumental variables. Secondly, the database as a whole shall statistically represent the variability of the observations. Finally, to ensure successful inversions, there should be a sufficiently large number of database cases to ensure that each observation matches multiple states. Therefore, a high number of rigorous all-sky simulations must be performed.

To capture the global and annual variability of observations, coverage was chosen to correspond to CloudSat overpasses from 2009 and 2010. ERA5 was selected as ancillary data to provide atmospheric, meteorological, and surface information for these overpasses. Further details are provided in Sect. 3.2.

The basic strategy applied for the generation of the preliminary database was found consistent with the requirements and was therefore kept. However, several extensions were required for the new database. Therefore, a full reimplementation of the simulation environment was necessary in order to increase the calculation efficiency in such way that the extensions were feasible. In particular, a simplified description of the sensor was used for the generation of the preliminary database. Incorporating a more complete description of the antenna and channel responses in the new database results in a significant increase in radiative transfer calculations.

A second main area of improvement was to extend the description of ice hydrometeor properties to include a non-uniform probability of occurrence and a computationally efficient, approximate treatment of particle orientation. The latter was required

to cover the polarisation response of ICI's channels, which was not accounted for in the preliminary database. Remaining improvements include considering the full interference of ozone in the atmospheric radiative transfer calculations, a better description of the emissivity of snow-covered surfaces, and a small correction originating in the calibration procedure.

The omission of a full antenna pattern has been a significant limitation in all previous attempts to develop retrieval databases. If an observation is represented with a single pencil beam calculation, the decrease in radiance caused by the presence of ice hydrometeors tends to be overestimated (Barlakas and Eriksson, 2020). Accordingly, the 'beam-filling' must be captured (Davis et al., 2007). By including the along-track antenna response, the preliminary database went further than most earlier works. However, incorporating a two-dimensional antenna response provided the largest challenge in the extension of the framework. This challenge had several parts. Firstly, it raised the question of how to generate three-dimensional atmospheric scenes based on CloudSat data that have only a coverage of two-dimensional character (height and along-track). This was solved by implementing the method of Barker et al. (2011).

The next consideration is how to perform the radiative transfer calculations. Full three-dimensional calculations are infeasible due to the associated high computational burden. Instead, an independent beam approximation (IBA) was selected. This entails sampling the atmosphere along a number of (slanted) directions and performing local one-dimensional radiative calculations. This is followed by a weighting of the obtained pencil beam brightness temperatures with the antenna response to obtain the antenna temperature for each frequency. Barlakas and Eriksson (2020) showed that this IBA approach, when applied within the frequency range of ICI, removes the simulation bias typically present for single pencil beam calculations, albeit with a smaller random error remaining. More details are found in Sect. 3.5.

A summary of the inputs, process and outputs of the simulation environment is given in Fig. 1. The different parts of the framework are presented in the subsequent sections. For further details, the reader may refer to the study report (Rydberg et al., 2023). The content of the database delivered to EUMETSAT is found in Table 1. The core variables of the database are IWP, $Z_{\mathrm{m}}$ and $D_{\mathrm{m}}$, and corresponding simulated ICI observations in the form of antenna temperatures $T_{\mathrm{a}}$. The atmospheric variables are antenna-weighted values and there is an assumption of remapped ICI data to a common footprint (see Eriksson et al. (2020)). An extended database containing, for example, the profiles of ice water content underlying the IWP values, has been saved to be used for future research purposes.

## 3.2 Generation of 3D hydrometeor fields

Cloud radars are currently the best source of information on vertical cloud structure. The 94 GHz Cloud Profiling Radar (CPR) aboard CloudSat provides measurements in the form of radar reflectivities at 500 m vertical resolution, offering nearly global coverage. CloudSat data from 2009 and 2010 were used, avoiding the fact that CloudSat offers only daytime coverage since 2011. However, CloudSat offers only two-dimensional data, lacking information in the across-track direction. To expand into a three-dimensional representation, and therefore cover horizontal cloud structure, the 3D cloud-construction algorithm of Barker et al. (2011) was implemented. In the algorithm, a three-dimensional cloud structure is generated through the colocation of passive satellite and two-dimensional radar data. Both CloudSat and The Moderate Resolution Imaging Spectroradiometer

**Table 1.** Variables included in the ICI cloud radiation retrieval database to be used operationally at EUMETSAT. Ice water path refers to the *total* ice water path, i.e. both cloud ice water path and precipitating ice.

| Variable | Unit | Description |
|---|---|---|
| $T_{\mathrm{a,cs,i}}$ | K | Simulated clear sky antenna temperature for ICI channel $i$. |
| $T_{\mathrm{a,as,i}}$ | K | Simulated all sky antenna temperature for ICI channel $i$. |
| $\Delta T_{\mathrm{a,i}}$ | K | Simulated cloud signals in antenna temperature (Eq. 3). |
| $\tau_i$ | - | Cloud optical depth for ICI channel $i$. |
| IWP | $\mathrm{kg\,m^{-2}}$ | Ice water path (total). |
| $Z_{\mathrm{m}}$ | m | Mean ice hydrometeor height by mass. |
| $D_{\mathrm{m}}$ | m | Mean particle size by mass, in terms of a mass equivalent spherical diameter. |
| RWP | $\mathrm{kg\,m^{-2}}$ | Rain water path. |
| LWP | $\mathrm{kg\,m^{-2}}$ | Liquid water path. |
| TCWV | $\mathrm{kg\,m^{-2}}$ | Column water vapour. |
| Surface Type | - | Possible types: ocean, land, snow, sea ice, mixed. |
| Surface pressure | Pa | - |
| Surface temperature | K | - |
| Surface wind | $\mathrm{m\,s^{-1}}$ | Surface wind speed. |
| Latitude | degrees | - |
| Longitude | degrees | - |
| Ice habit | - | Frozen hydrometeor model, incorporating habit and PSD. |

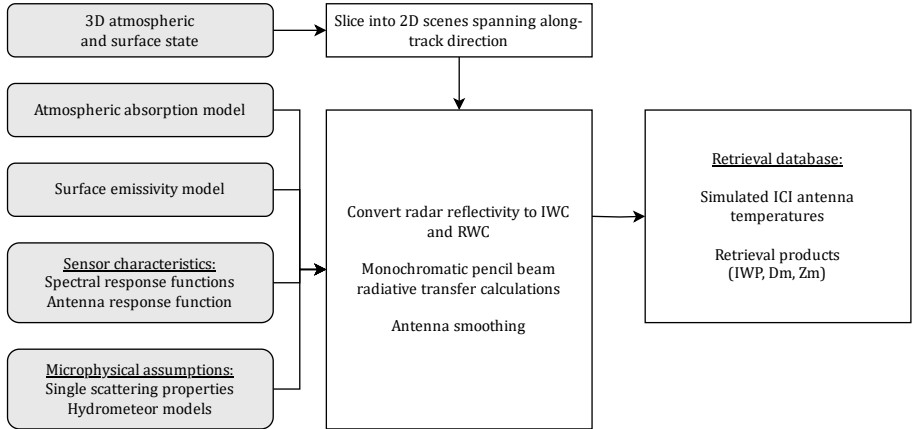

**Figure 1.** Summary of the inputs, processes and outputs of the retrieval database generation scheme.

**Table 2.** Particle models used within simulations of the ICI retrieval database, given alongside habit and particle size distribution (PSD). The source for the PSD is given in each case. The aARO scaling factor is randomly chosen within the given range. Also given is the occurrence fraction, or the probability that a given model is randomly selected for each atmospheric scene, denoted as $p_i$

| Model | Habit | PSD | aARO factor | $p_i$ |
|---|---|---|---|---|
| AA1 | Large plate aggregate | F07 - Tropics | 1 - 1.6 | 0.3 |
| AA2 | Large column aggregate | F07 - Tropics | 1 - 1.6 | 0.1 |
| AA3 | Large block aggregate | D14 | 1 - 1.6 | 0.13 |
| IWC | Six bullet rosette | D14 | 1 - 1.6 | 0.2 |
| Snow | Evans snow aggregate | F07 - Midlatitude | 1.4 - 1.6 | 0.1 |
| Graupel | Eight column aggregate | D14 | 1 - 1.2 | 0.17 |

(MODIS) onboard Aqua are part of NASA'S A-train constellation, and therefore collocated data from the two instruments

exist. The Barker algorithm was applied directly to CloudSat radar reflectivities, avoiding the use of retrieved data.

The three-dimensional cloud structures obtained using the Barker algorithm are then merged with ERA5 data to provide a full description of atmospheric gases, meteorological conditions and the surface type within the scene. The output is a three-dimensional scene containing all relevant atmospheric and surface parameters.

Finally, the three-dimensional distributions of radar reflectivities are converted to fields of hydrometeor contents. The rela-

tionship between radar reflectivity and IWC is conditioned on microphysical assumptions (Kulie et al., 2010; Ekelund et al., 2020). The reflectivities, based on the microphysics described in Sect. 3.3, are calculated on a grid of IWC, leading to a table of IWC-dBZ values and associated attenuation. The inversion of radar reflectivities within a given scene to IWC fields makes use of the look-up table, following the setup described in Sect. 3.2 of Ekelund et al. (2020). The same approach is used for the generation of rain water content for temperatures above $0°C$. Liquid water content (LWC) is taken from ERA5, since CloudSat

is largely insensitive to non-precipitating water. The final scenes each span approximately $2000 \, \text{km} \times 50 \, \text{km}$ in the along and across-track direction.

## 3.3  Ice hydrometeor particle models

At microwave and sub-millimetre wavelengths, scattering by atmospheric ice has a significant effect on measured radiances. Radiative transfer simulations for ICI therefore require assumptions on the optical properties of frozen hydrometeors. Six dis-

tinct particle models were implemented for this purpose, where the term particle model refers to a particle habit, an associated particle size distribution, and a range of scaling factors to imitate orientation effects. The six particle models are shown in Table 2. A greater number of particle models are now used in simulations for the new database than used for the preliminary database.

Since scattering calculations of ice hydrometeors are computationally expensive, it is preferable that the single scattering properties are pre-calculated. Additionally, ice hydrometeors are highly variable in shape (O'Shea et al., 2016). Taking these factors into consideration, the chosen habits reflect a sample of the standard habits present in the ARTS single scattering database (Eriksson et al., 2018), which offers single scattering data at microwave and sub-millimetre wavelengths for hydrometeors ranging from simple liquid spheres to more advanced ice aggregates.

The three particle models denoted with 'AA' in Table 2 each use a mixture of two distinct habits: one for smaller particles and one for larger particles. To represent large particles, aggregates are used. The smaller particle habits are single crystals that correspond in shape to the aggregate they are paired with. For example, the large plate aggregate habit is complemented with the thick plate crystal. These three particle models are intended to be generic, exhibiting different levels of extinction (Ekelund and Eriksson, 2019). The name used to represent the habit mixture under 'Habit' in Table 2 refers to the name of the large particle habit, i.e. the ARTS aggregate (AA) habit. The remaining three habits used are an attempt to cover specific hydrometeor categories, e.g. graupel.

Particle size distributions (PSD) are sourced from Field et al. (2007) (represented as F07 and available for both tropics and mid-latitudes) and Delanoë et al. (2014) (represented as D14). The inclusion of D14 is a new addition to the database. A single moment version is used, with the coefficients for the modified gamma taken from Table 4 in Delanoë et al. (2014), under 'All (DARDAR)'. The parameterisation for $N_0^*(T)$, the intercept parameter of the normalised size distribution, follows 'DARDAR' in Table 5 of Delanoë et al. (2014).

The assumption of totally random orientation (TRO) of frozen hydrometeors is frequently made in radiative transfer simulations of microwave observations. In reality, microwave imagers observe polarisation differences, i.e. differences in measured brightness temperatures between vertical (V) and horizontal (H) polarisations (Defer et al., 2014; Gong and Wu, 2017). This is consistent with the presence of oriented ice particles. However, the small simulated polarisation differences that arise when using TRO are far from realistic. Therefore, we implement an approximation of azimuthally random orientation (aARO) according to the scheme developed by Barlakas et al. (2021) and extended by Kaur et al. (2022).

In the scheme, azimuthally random orientation effects are mimicked by scaling the extinction values obtained from the TRO assumption. This is achieved with a factor by which extinction at V- and H-polarisation are weakened and strengthened, respectively. The factor is equal to 1.0 for TRO, and a higher factor increases the difference between V- and H-polarisation. The factor for a given simulation is randomly selected from within a pre-determined range, denoted as the aARO factor in Table 2. Kaur et al. (2022) demonstrated that the best performance at microwave wavelengths (166 GHz) was achieved when using a uniform distribution between 1.0 and 1.4, although the result was somewhat dependent on microphysics assumptions and thus may be adjusted according to the choice of habit. To account for the presence of sub-millimetre wavelengths in ICI and thus potentially larger polarisation differences, the distributions were extended to 1.6 for most particle models. Additionally, Barlakas et al. (2022b) demonstrated that there exists asymmetry between the polarisations, and therefore a larger scaling was applied to H-polarisation than at V-polarisation.

Particle habit and PSD can both depend strongly on environmental conditions and cloud-type (Yau and Rogers, 1989) and therefore do not occur at the same rate (O'Shea et al., 2016). Additionally, the parameters describing the assumed particle

model (including the aARO factor) can have a strong impact on the mapping of IWC from reflectivity inversions (Ekelund and Eriksson, 2019). In light of these considerations, the six particle models are not used with equal frequency within the simulations. Instead, the selection of particle model to use within the simulations in a given scene is a weighted random selection process, where the probability of selecting a particle model follows a non-uniform probability distribution. To assign the selection probabilities, a batch of reflectivity-IWC inversions were performed for each of the particle models. An overall distribution of IWC at an altitude of 8 km was constructed by weighting the IWC distributions from each individual particle model at the same altitude. The weights were chosen according to three criteria. Firstly, that each particle model occurs in a minimum of 10% of the database cases. Secondly, that the particle model 'AA1' occurs in a minimum of 30% of the cases, motivated by its strong performance in previous studies (Ekelund et al., 2020; Pfreundschuh et al., 2020; Geer et al., 2021; Kim et al., 2024). Finally, the weighted IWC distribution should agree with an the true distribution of IWC, so far as the previous two criteria allow.

In the absence of 'true' IWC data, satellite observations are used. The DARDAR (raDAR/liDAR) product offers reliable retrievals of IWC (Cazenave et al., 2019). The product is derived from the 95 GHz Cloud Profiling Radar (CPR), carried on CloudSat, and the 532 and 1064 nm lidar CALIOP (Cloud-Aerosol Lidar with Orthogonal Polarization) lidar, carried on CALIPSO (Cloud-Aerosol Lidar and Infrared Pathfinder Satellite Observation). The observed distribution used for our analysis was calculated using the CloudSat-based product DARDAR v3.1.

## 3.4 Radiative transfer

This section outlines the calculation of monochromatic pencil beam brightness temperatures. The weighting of these with sensor response data is discussed in the subsequent section. The radiative transfer calculations were performed using the Atmospheric Radiative Transfer Simulator (ARTS) (Buehler et al., 2018).

The static forward model input includes absorption data. The data are taken from the following sources: Nitrogen (Liebe, 1993), oxygen (Rosenkranz, 1993), water vapour (Mlawer et al., 2012), and ozone (Pickett et al., 1998). The choice of absorption models for oxygen and water vapour followed recommendations of an EUMETSAT study updating and developing RTTOV for sub-millimetre (Fox et al., 2024).

At frequencies below 325 GHz, surface contributions can be non-negligible and the surface emissivity must be considered. Over ocean and land, simulations use the Tool to Estimate Sea-Surface Emissivity from Microwaves to sub-millimetre waves (TESSEM[2]) (Prigent et al., 2017) and the Tool to Estimate Land-Surface Emissivities at Microwave Frequencies (TELSEM[2]) (Wang et al., 2017). TELSEM[2] provides full emissivity parameterisations up to 85 GHz for all land and sea-ice surfaces, except for new and first-year ice where an extension up to 183 GHz is made. Above these frequencies, constant surface emissivities are generally assumed. However, Harlow and Essery (2012) observed increasing emissivity with frequency for stratified snow and decreasing emissivity with frequency for new snow. Additionally, at higher frequencies, Wang et al. (2017) found some disagreement between retrieved sea-ice emissivities from ISMAR observations and TELSEM[2] sea-ice emissivities. Therefore, we developed an empirically-based probabilistic model for snow and sea-ice surface types. Emissivities at both V and H polarisations are generated from a multivariate Gaussian distribution. The ranges of the distributions were determined using

emissivities from Hewison et al. (2002); Harlow (2009); Harlow and Essery (2012). The means, variances, and correlation matrices of the distributions are built on emissivity retrievals performed in Munchak et al. (2020). At frequencies 183 GHz and greater, we use Risse (2021) for reference values. Beyond 243 GHz, constant emissivities are assumed.

It was judged too computationally expensive to perform all calculations with a full scattering solver. Instead, two sets of monochromatic pencil beam calculations were performed. Clear-sky radiances were calculated with nominal humidity or with relative humidity fixed to 50%. These calculations excluded all hydrometeors and were performed for multiple frequencies inside each of ICI's passbands.

All-sky simulations were made with ARTS' interface to the DISORT (Discrete Ordinate Radiative Transfer) scattering solver, at one frequency per passband. The DISORT algorithm uses the discrete ordinate method to solve the radiative transfer equation in a multi-layered medium, handling the absorption, emission, multiple scattering, and lower-boundary reflection of monochromatic radiation (Stamnes et al., 1988). For reference, a clear-sky DISORT run was also made. How these calculations were combined to obtain final values is discussed in the following section.

## 3.5 Sensor characteristics

Radiances observed by the ICI antenna depend on the frequency response and the angular response of the sensor, and both of these factors must be taken into consideration.

Measured spectral response functions vary within the channel passbands. Therefore, if a constant spectral response function is applied, a bias in brightness temperature may occur. The effect was seen to be strongest for channels 06V (325.15±3.50 GHz) and 09V (448.00±7.20 GHz). To avoid such bias, the full spectral response functions, based on measured ICI data, are now included for all channels.

As briefly discussed in Sect. 3.4, clear-sky calculations are performed for multiple frequencies. The spectral radiances must be obtained on a frequency grid with a resolution that is fine enough to capture variation of the spectral response function over the passband (resolution of 10 MHz). To lessen computational load, simulations are run for a reduced number of frequencies across the sideband and then interpolated onto the finer grid. The number of frequencies per sideband to be simulated was allowed to vary between channels. The number for each channel was chosen such that the resulting uncertainties contribute less than 5% of NE$\Delta$T. The number needed to fulfil this criterion largely depends on the contamination of ozone at the given frequency, e.g. the 664 GHz channel requires the highest amount at 25 frequencies per sideband. Due to the high computational cost, all-sky simulations are performed using only the centre frequency of the two sidebands.

After application of the spectral response function, spectral radiances are converted to a brightness temperature for a given channel using the inverse Planck function. The conversion of spectral radiances to brightness temperatures can be well approximated by the linear relationship:

$$T_{\mathrm{b}} = \frac{B^{-1}(f_{\mathrm{centre}}, R_{\mathrm{i}}) - b_{\mathrm{i}}}{a_{\mathrm{i}}}. \tag{1}$$

$B^{-1}$ is the inverse Planck function, $R_i$ is the spectral radiance for channel $i$, and $f_{\text{centre}}$ is the nominal centre frequency of the channel. $a_i$ and $b_i$ are band-correction coefficients that are derived pre-launch, based on the characteristics of the channel spectral response function. The coefficients were provided by EUMETSAT.

The incorporation of the full two-dimensional antenna response is necessary in order to avoid beam-filling errors. This is a particularly important consideration for ICI, due to the relatively large footprint size. To perform the antenna weighting, data are integrated over the sensor field of view:

$$T_a = \int_{\Omega} T_b(\Omega)G(\Omega)d\Omega. \tag{2}$$

$G(\Omega)$ is the normalised antenna gain function, provided by EUMETSAT, and $\Omega$ is the solid angle. The pattern associated with the 183 GHz channel is applied to all channels, to be consistent with the remapping of data within the operational algorithm. $T_a$ is a simulated antenna temperature that matches a remapped observation on the sub-point of the satellite orbit.

To accurately estimate $T_a$, the $T_b$ field over the antenna footprint should be relatively dense. However, achieving this with purely simulations would not be computationally feasible. This was solved by using fewer pencil beam simulations, arranged in a pre-determined configuration within a footprint. The simulations are then interpolated onto a finer grid. Within the convex hull of the simulations, a linear interpolation is performed. Outside this region, a nearest neighbour extrapolation is applied.

The configuration of pencil beams will have an impact on the accuracy of antenna weighted data, but a high number of simulations is undesirable. To achieve better efficiency of the ARTS calculations, the three-dimensional scenes were divided into parallel two-dimensional slices oriented in the along-track direction. As a consequence, the arrangement of pencil beams consists of a number of tracks corresponding to the two-dimensional slices. In order to determine a configuration that gives a reasonably low final error, a number of combinations of tracks and spacing along-track were tested. Once simulations were performed for a given configuration, the interpolation onto the finer grid was performed and Equation 2 was applied. To provide a reference value for the error calculations, $T_a$ was also calculated using simulations performed for each point on the finer grid.

The chosen configuration used simulations spaced approximately 4.4 km apart along the centre track, i.e. the satellite sub-point. This spacing was successively doubled for each track away from the centre. A total of 38 pencil beam simulations spaced across 11 tracks are used to represent the spatial variation of brightness temperatures in each final antenna temperature. Overall, the reduction in the number of simulations achieved approximately zero bias and a standard deviation of 0.5 K. This standard deviation is an average over all cases tested, and is generally low for clear-sky cases but increases with hydrometeor impact.

To compute the $T_b$ field at a specific location, a target position is selected. The position is located along the centre track, imitating the boresight of the antenna directed towards the target. Targets were selected such that the along-track distance between the database $T_a$ values is approximately 8.9 km. As a result, there is some overlap in the antenna patterns used to calculate each $T_a$. Upon selection of a target, the sensor position must be calculated. A challenge then arises when deciding on which incidence angle to use. While all channels produce a common footprint pattern, their instantaneous footprints at surface level are not located at the same position due to variation in incidence angle (Table 1 of Eriksson et al. (2020)). To address this, the same ground-level footprint was assumed for all channels. The position of the sensor was then adjusted based on the relevant incidence angle, accommodating the variation between channels.

The re-gridding of brightness temperatures is performed onto an area bounded by ($\pm0.7°$, $\pm0.8°$) in zenith and azimuthal angle relative to the sensor line of sight, respectively. This is approximately equivalent to twice the full width at half maximum (FWHM) of the antenna pattern. Each set of clear-sky and all-sky simulations are weighted with the antenna response function separately.

The measurement vector, to be used in the operational L2 retrieval for ICI, consists of cloud signals for each ICI channel. The cloud signal is obtained by subtracting a clear-sky simulation from the observed antenna temperature. The clear-sky antenna temperature is simulated using RTTOV, incorporating geophysical data from ECMWF. The data includes atmospheric temperature, surface wind speed, ozone, and surface characteristics. Since humidity data from ECMWF can be unreliable, a fixed relative humidity is used within the simulations. Further details on the geophysical data used, and the humidity profile in particular, can be found in Sect. 3.4.3 of Eriksson et al. (2020).

The objective for generating a cloud signal is to decrease the contribution from the background. In the case of the database, a cloud signal can be obtained from the difference between an all-sky simulation and an all-sky simulation excluding hydrometeors, i.e. the clear-sky DISORT run introduced in Sect. 3.4. This value can be viewed as a 'true' cloud signal, but there is a risk that uncertainties arising from non-perfect humidity data are introduced. To adjust for the inclusion of ECMWF data and be consistent with the measured ICI cloud signal, the quantity $\Delta T_\mathrm{a}$ is introduced. Within $\Delta T_\mathrm{a}$, the two clear-sky runs are also included as follows:

$$\Delta T_\mathrm{a} = T_\mathrm{a,cs} + T_\mathrm{a,as} - T_\mathrm{a,as,no-hm} - T_\mathrm{a,cs,fixed-rh}. \tag{3}$$

$T_\mathrm{a,cs}$ refers to a clear-sky antenna temperature, $T_\mathrm{a,as}$ to all-sky, $T_\mathrm{a,as,no-hm}$ to a simulation with no hydrometeors included, and $T_\mathrm{a,cs,fixed-rh}$ to a simulation with fixed relative humidity.

Overall, passband properties are captured well in clear-sky simulations. An increasing amount of scattering in simulations (i.e. higher $\Delta T_\mathrm{a}$) leads to progressively worse representation of the passband and thus additional simulation uncertainty. However, this uncertainty is small compared to uncertainties arising from other aspects of the simulation, such as hydrometeor assumptions and the neglect of some three-dimensional effects (Barlakas and Eriksson, 2020).

### 3.6 Output

The database consists of $\sim 9.4 \times 10^6$ cases, where each case is a set of simulated ICI radiances within a remapped footprint, accompanied by aforementioned cloud ice variables. Simulations were performed using $5 \times 10^4$ atmospheric scenes. Each database case consists of the quantities given in Table 1. An example of simulation outputs for a given swath is shown in Fig. 2.

### 4 Retrieval approach

The retrieval of atmospheric quantities is an ill-posed problem in the sense that for a given observation there may be more than one solution describing the underlying atmospheric state, due to the associated uncertainties. Observations uncertainties are

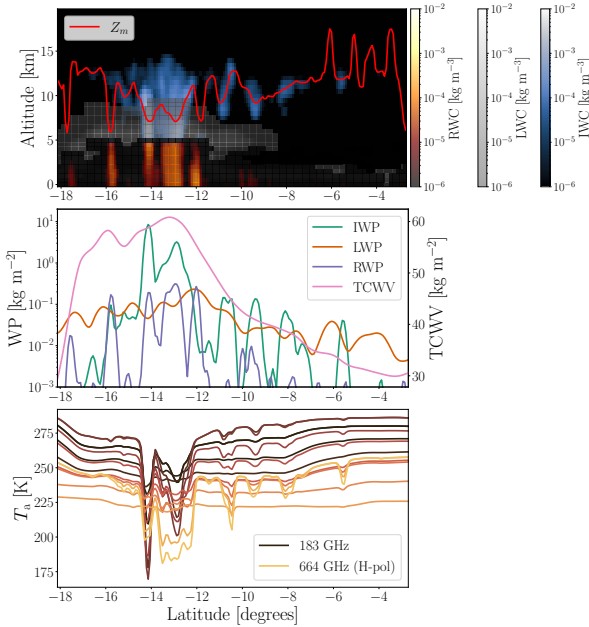

**Figure 2.** Representation of simulation outputs across the flight direction of CloudSat. IWC, RWC and LWC (top) are not included in the cloud radiation retrieval database to be used operationally, but calculated as an intermediate step. Also shown are $Z_m$, IWP, LWP, RWP, TCWV, and $T_a$ for all ICI channels, where Table 1 can be referred to for variable definitions. $T_a$ lines are plotted with a gradient colour, where increasing lightness of colour indicates increasing frequency.

generally known only statistically, implying that a retrieval method that retrieves only a single quantity from an observation is unsuitable for the problem at hand. Instead, it is more appropriate that the retrieval is given in the form of a probability density function (PDF) (Rodgers, 2000).

A Bayesian approach is the obvious method if one wishes to perform probabilistic retrievals. Within a Bayesian framework, the solution to the inverse problem is provided in the form of a posterior distribution $p(x|y)$, representing the distribution of possible states conditional on the observation. Since the posterior distribution represents our knowledge of the retrieved quantity after taking into account observations, prior beliefs, and measurement and model uncertainties, it offers a natural way to characterise the uncertainty of the retrieval.

Commonly used inversion methods such as the optimal estimation method (OEM) do adopt a Bayesian framework, albeit a simplified version that produces an approximately Gaussian posterior. However, this becomes unsuitable in the presence of the non-Gaussian statistics and the non-linearity of the forward model that often characterise the retrieval problem (Pfreundschuh et al., 2018).

Bayesian Monte Carlo integration was shown to be successful in the context of ICI retrievals (Eriksson et al., 2020) and will be used in ICI-side of the L2 product at EUMETSAT. However, after the operational algorithm was finalised, quantile regression neural networks (QRNN) began to emerge as an alternative. QRNNs have been shown to perform successful probabilistic

retrievals (Pfreundschuh et al., 2018) in the sense that they allow for the estimation of a distribution rather than a single point estimate. Several studies have successfully implemented QRNNs in the retrieval of cloud ice products (Amell et al., 2022). The method also has the potential to perform inversions of an entire swath, since it can make use of the spatial information from observations within the area of interest (Pfreundschuh et al., 2022a). There are other retrieval approaches that allow for uncertainty quantification. One such approach is to use a Bayesian Neural Network (BNN). BNNs have the strength of being able to estimate both aleatoric and epistemic uncertainty, and have been successfully used to retrieve of IWP and $D_\mathrm{m}$ (Dong et al., 2023). However, Pfreundschuh (2022) observes that aleatoric uncertainty is often the dominant source of uncertainty in the retrieval problem, which QRNN are suited to handle. Furthermore, BNNs take more time to train and evaluate. Since QRNN has established itself as a faster and more flexible approach to the retrieval problem, it became the obvious choice for our test ICI retrievals. A description of the QRNN approach is given in Appendix A, and details on the configuration of the model are provided in Appendix A1.

## 5  Results and discussion

### 5.1  Simulation of other instruments

The success of the inversion method is highly reliant on the simulation quality of the database. Due to the lack of real ICI observations, assessing whether the simulations are consistent with reality becomes challenging. The approach taken to address this difficulty was to conduct simulations of similar, existing instruments, namely The Global Precipitation Measurement (GPM) Microwave Imager (GMI), The Microwave Airborne Radiometer Scanning System (MARSS) and The International Submillimetre Airborne Radiometer (ISMAR). The simulated radiances were subsequently compared to real observational data. Since actual flight paths were not simulated and the time period considered differs between simulation and observation, the comparisons were made in a statistical sense. Simulation of GMI allows for a comparison of global simulations, whereas simulation of ISMAR and MARSS functions as a regional study with the advantage of covering sub-millimetre wavelengths.

### 5.1.1  Simulation of GMI

The GMI instrument is a satellite-borne conically scanning microwave radiometer operating at frequencies ranging from 10 GHz to 183 GHz. We simulated observations from the four high-frequency GMI channels that measure at 166 GHz and 183 GHz. The two 183 GHz channels were chosen due to the benefit of the overlap with the ICI 183 GHz channels. The two 166 GHz channels measure at V and H polarisation and thus offer the opportunity to test the improvements to the simulation framework with regard to polarisation and particle orientation. The simulations used GMI sensor characteristics, i.e. incidence angle, spectral response function and footprint size. To account for GMI instrument error, noise was added to the simulated $T_\mathrm{a}$ in the form of a zero-mean Gaussian with standard deviation according to Table 1 in Kaur et al. (2022). Since GMI has only been operational from 2014, simulations are compared to GMI observations from 2020.

Both the one- and two-dimensional distributions of $T_\mathrm{a}$ show good agreement between observations and simulations, although a few small discrepancies are present (Fig. 3). In the one-dimensional distributions, the peak at high $T_\mathrm{a}$ corresponds to clear-sky cases. There is strong agreement in the peaks. However, the observations reach slightly higher temperatures in the case of the 166 GHz channels. It was found that these cases occurred in desert regions. The reason for this difference is likely that the emissivities assumed in the simulations are too low. TELSEM land emissivities are derived only for 85 GHz, and it is not known if the extension to 166 GHz remains valid. This highlights the need for improved estimates of emissivities at higher frequencies. A second potential reason is that the ERA5 skin temperatures used in the simulations are too low in desert regions.

Good agreement is also seen for most cases with cloud impact (lower $T_\mathrm{a}$). There is some disagreement at the lowest temperatures, where it can be seen that cases of $T_\mathrm{a} < 100$ K are present in the observations and not in the simulations. However, these observed cases occur below a probability density of $10^{-6}$ K$^{-1}$. Since the simulation distributions are computed with only $\sim 7 \times 10^6$ cases, it is unsurprising that scenarios of $T_\mathrm{a} < 100$ K are not well represented in the simulations.

The aim of simulating GMI is not necessarily to achieve perfect agreement with observations, since we do not simulate the same time frame and flight paths that the observations cover. Rather, it is important that the simulations fully cover the variability of the observations. Aside from the region of $T_\mathrm{a} < 100$ K and $T_\mathrm{a} > 300$ K, this holds true.

Although cloudy simulations do agree well with observations, there is room for improvement. One factor to consider is the particle models, which could be fine-tuned to better capture the behaviour of the observations. For example, the particle model could be selected based on the cloud-type, as would be the case in reality. However, we will wait until real ICI observations to make adjustments to the simulation set-up, including the particle models. This allows us to compare the distribution of $T_\mathrm{a}$ across all 13 ICI channels, providing a more comprehensive validation.

When comparing the joint distributions, simulations and observations are in good agreement on both the extent of the correlations and the location of the contour levels. Again, the observed distributions extend to lower temperatures than the simulations, which is expected due to fewer simulations in total. There is a clear difference between the 166V and 166H channel $T_\mathrm{a}$. This implies that the introduction of the approximation of azimuthally oriented particles in the new database produces the required effect in the simulations. To further check the agreement on polarisation effects, the polarisation difference was plotted (not shown) against the 166V channel $T_\mathrm{a}$, across a range of surface types. The resulting figures were very similar to Fig. 4 in Kaur et al. (2022). Likewise, good agreement between the simulations and observations was found, confirming that the simulations capture the variability in polarisation difference in the case of the 166V and 166H channels.

### 5.1.2 Simulation of ISMAR and MARSS

ISMAR is a sub-millimetre radiometer that operated on flight campaigns that serve as airborne demonstrators for ICI (Fox et al., 2017). Also on board the aircraft were MARSS radiometers (McGrath and Hewison, 2001), providing measurements at microwave wavelengths. Together, the observations cover the range of ICI channel frequencies. In the case of the 325.15±1.5 GHz channel, only observations prior to 2019 were used due to issues with later flights.

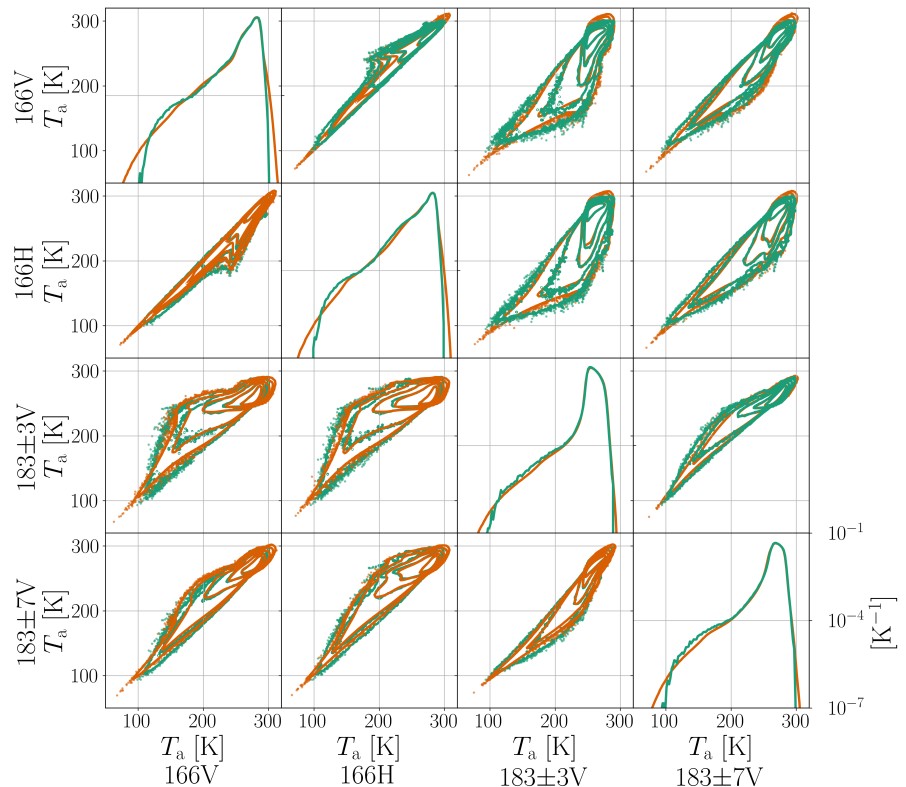

**Figure 3.** Distributions of simulated (green) and observed (orange) antenna temperatures $T_a$ for four high-frequency GMI channels, simulated using the database generation framework developed for ICI. The plots in the upper right triangle are identical to those on the lower left triangle, but simply have the simulations plotted over the observations, rather than under. Contour lines in the joint distributions (off-diagonal) correspond to $(10^{-1}, 10^{-2}, 10^{-3}, 10^{-4}, 10^{-5}, 10^{-6}, 10^{-7})$ K$^{-2}$. Samples that occur in $T_a$ bins with a probability density of less than $10^{-7}$ K$^{-2}$ are represented by the scatter points in the joint distributions. GMI observations are taken from the year 2020. The simulations and observations both lie within the latitude range of $[-60°, 60°.]$

Individual flights were not simulated. Instead, simulations were performed within a subset of the three-dimensional scenes used for creating the ICI retrieval database (similarly to the GMI simulations). The location of the simulations and the observations used is shown in Fig. 4. Noise was added to simulations according to Table 3 in Fox et al. (2017).

Similarly to the GMI simulations, the aim of this comparison is to verify that the simulated data spans the measurement space of real observations. Generally, the variability of brightness temperatures present on the flight campaigns is well captured by the simulations (Fig. 5). There are a small number of cases where observations lie outside the simulations. However, it is likely

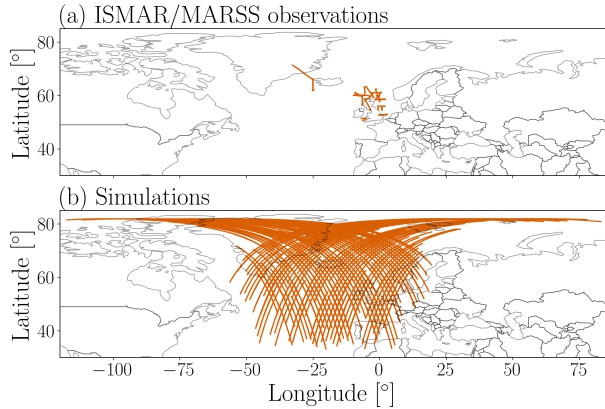

**Figure 4.** Locations of observations from the ISMAR and MARSS flight campaigns are displayed in panel (a), using cases where flight altitude was greater than 9 km. Locations of the simulated ISMAR and MARSS observations are displayed in panel (b).

that some observations suffer from higher noise than specified in Fox et al. (2017), possibly explaining the discrepancies. This is most evident in the distributions of the channel at 325.15±3.50 GHz.

Similar features are present in both the simulations and the observations. For example, there are two distinct 'arms' of data visible in many of the plots in Fig. 5. In the case of channels with low surface sensitivity, the two arms correspond to clear-sky and cloudy cases. This can be seen clearly in the joint distribution of the 183.25±3.00 GHz channel and the 325.15±9.50 GHz channel. The clear-sky cases follow the identity line. The arm that diverges from the identity line is a product of scattering due to ice hydrometeors. It extends to increasingly colder $T_\mathrm{a}$ as the difference between channel frequencies increases. Such an effect is attributed to the scattering strength increasing with frequency. However, this is not always the case; the channel at 183.25±7.0 GHz produces lower temperatures relative to the channel at 325.15±3.50. This is due to the lower frequency channel having higher atmospheric transmission than the high frequency channel and therefore interaction with a larger fraction of the column of hydrometeor ice.

In channels sensitive to the surface, the arm arises from the surface contribution. The effect is most evident in channels at the same frequency but different polarisations, such as for the two channels at 243.20±2.50 GHz, where a significant polarisation difference can be seen. This is more evident in the case of the simulations, where the polarisation difference reaches 50 K. Such high polarisation differences are attributed to dry conditions over the ocean, of which there are significantly more cases present in the global simulations than in the limited region covered by the flight campaigns, as shown in Fig. 4.

### 5.2 ICI Database radiances

One- and two-dimensional distributions of simulated $T_\mathrm{a}$ across all ICI channels are displayed in Fig. 6. A relatively large spread can be observed in the joint distributions between most channels, suggesting that all channels contain some independent information. It can also be noted that in the case of the 243 GHz channels and the 664 GHz channels, the two-dimensional distributions show a polarisation difference (the difference between vertically and horizontally polarised channels), which was

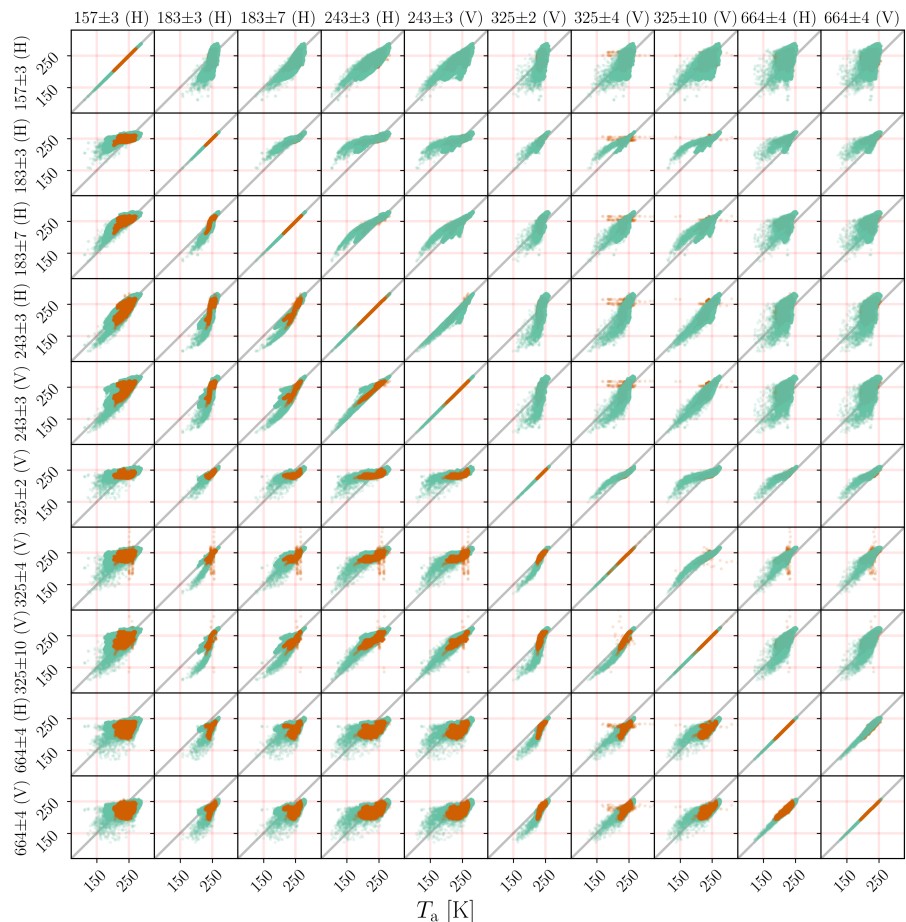

**Figure 5.** Comparison of observations (orange) from the ISMAR and MARSS flight campaigns and simulations (green) of ISMAR and MARSS using the database generation framework. The plots in the upper right triangle are identical to those on the lower left triangle, but simply have the simulations plotted over the observations, rather than under. The channels are labelled according to their centre frequency, with the corresponding sensor and the frequencies given to a higher precision found in B1.

absent in the preliminary database. The polarisation difference is displayed as a function of the vertically-polarised $T_a$ in Fig. 7. Horizontally polarised channels generally result in colder $T_a$, leading to predominantly positive polarisation differences. The 243 GHz channel is sensitive to the surface, and therefore polarisation differences due to both surface and cloud impact can be seen in panel (a) of Fig. 7. Polarisation differences arising from surface interactions (high $T_a$ cases) extend higher than cloud-impacted cases.

In the absence of the surface, polarisation difference depends on particle orientation, among other microphysical properties such as size and shape (Brath et al., 2020). We stress that the results shown here are simulations, not observations. Our simulations apply the same aARO factor and particle model across a given cloud column. This likely reduces the variation

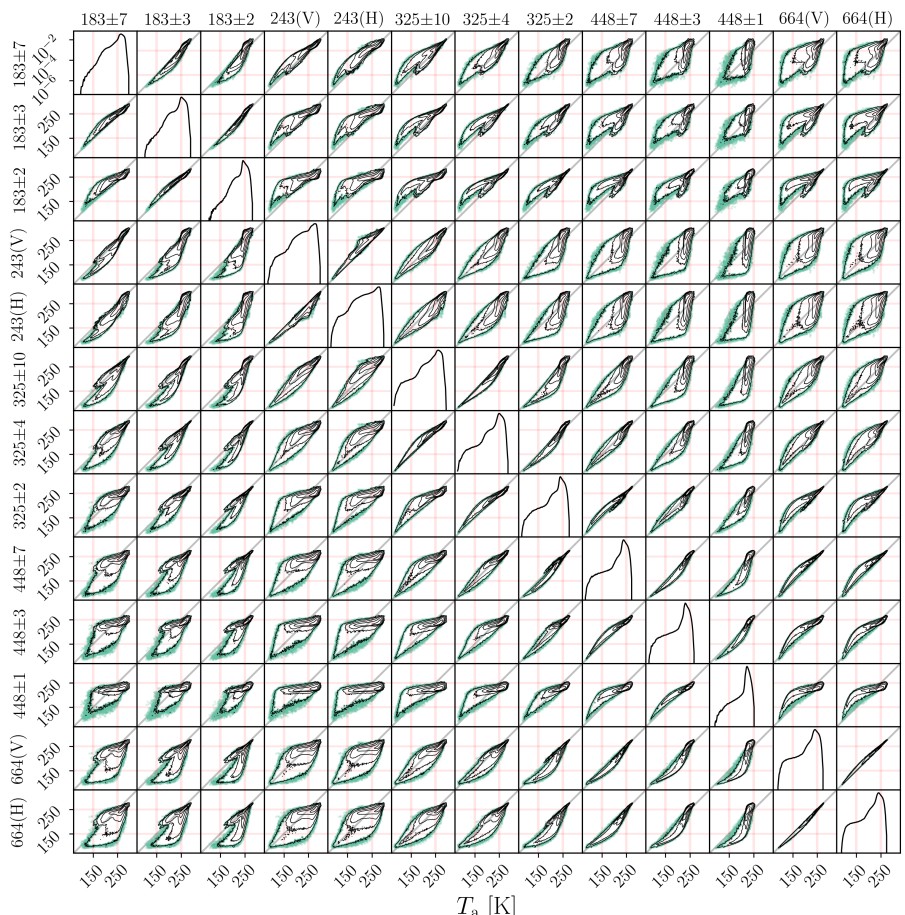

**Figure 6.** Distributions of simulated brightness temperatures for all ICI channels. ICI channels are labelled with their centre frequency, and the frequencies at higher precision can be found in Table 1 of Eriksson et al. (2020). When polarisation is not indicated in the label, the channel is V-polarised. Contour lines in the joint distributions (off-diagonal) correspond to $(10^{-1}, 10^{-2}, 10^{-3}, 10^{-4}, 10^{-5}, 10^{-6}, 10^{-7})$ $K^{-2}$. Samples that occur in $T_a$ bins with a probability density of less than $10^{-7}$ $K^{-2}$ are represented by the green scatter points in the joint distributions.

in polarisation difference between the channels, despite different sensitivities to altitude and particle size. Kaur et al. (2022) demonstrated that simulations performed at 166 GHz and 660 GHz, sampling across the same range of aARO factor and employing the same microphysics, led to a higher maximum polarisation difference at 166 GHz. If we extend the same conclusion to our ICI simulations, it is possible that a higher polarisation difference would be seen at 243 GHz when neglecting the surface, albeit small.

The features discussed in Sect. 5.1.2, i.e. the two distinct arms of data, are also present in the ICI simulations, confirming that the simulations are capable of emulating real physical behaviour.

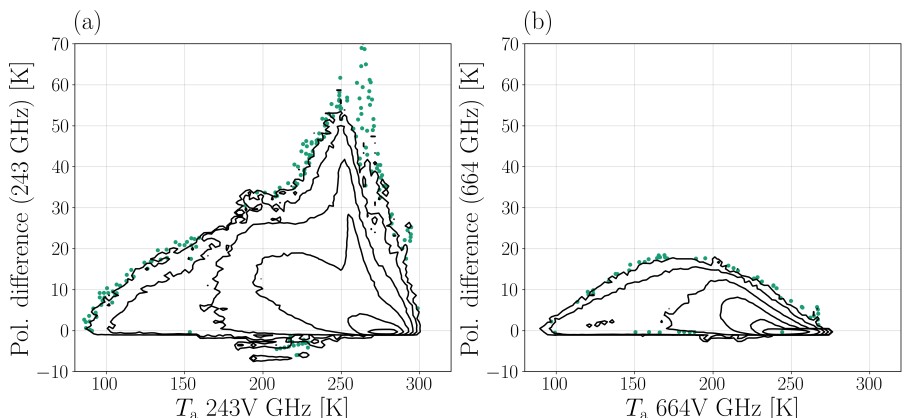

**Figure 7.** Polarisation differences as a function of $T_a$ at V-pol, plotted as contours of the two-dimensional normalised probability density function. The distribution at 243 GHz is shown in panel (a) and the distribution at 664 GHz is shown in panel (b). Contour lines correspond to $(10^{-1}, 10^{-2}, 10^{-3}, 10^{-4}, 10^{-5}, 10^{-6}, 10^{-7})$ K$^{-2}$. Samples that occur in bins with a probability density of less than $10^{-7}$ K$^{-2}$ are represented by the green scatter points.

### 5.2.1 Degrees of freedom

Retrievals will depend on the sensitivity of ICI observations to atmospheric conditions, above the instrument noise level. To
assess the sensitivity, we performed an analysis of the degrees of freedom (DoF) of the simulated ICI radiances. The DoF was calculated as a function of IWP and water vapour across all latitudes for both the new database and the preliminary database. The DoF quantifies the number of independent variables free to vary, and may be considered to be a measure of the information content of a measurement in the presence of noise.

    The DoF is essentially a comparison between the variance of the measurements and the variance of the measurement noise,
conducted in eigenspace. To achieve this, an eigendecomposition of the covariance matrix $\mathbf{S_y}$ of noise-free simulated $T_a$ is performed:

$$\mathbf{S_y} = \mathbf{E\Lambda E}^T, \tag{4}$$

where $\mathbf{E}$ is the matrix whose columns contain the eigenvectors of $\mathbf{S_y}$, and $\mathbf{\Lambda}$ is a diagonal matrix containing the eigenvalues. The diagonal matrix $\mathbf{S}_\epsilon$ is constructed to hold NE$\Delta$T$^2$ in its diagonal elements, and transformed to eigenvalue space:

$$\mathbf{S_\Lambda} = \mathbf{E S}_\epsilon \mathbf{E}^T. \tag{5}$$

The DoF can then be calculated as the number of eigenvalues that are larger than the corresponding diagonal element of $\mathbf{S_\Lambda}$. In the context of ICI, it can be interpreted as the effective number of channels.

    Panel (a) of Fig. 8 presents the DoF of the simulated ICI observations as a function of IWP and water vapour. The overall trend in DoF is consistent with Figure 8 in Eriksson et al. (2020), although the DoF is now calculated across all latitudes, rather

than only the tropics. An increase in either water vapour or IWP corresponds to an increase in DoF, although the change in DoF is much larger over the range of IWP. The general conclusion drawn from this result is that ICI is sensitive to humidity, but demonstrates strong sensitivity to ice hydrometeors.

We applied the same DoF analysis to both the new database and the preliminary database, and the resulting difference in DoF is presented in panel (b) of Fig. 8. Overall, there is an increase in DoF; a positive difference in DoF is present for all but very humid, clear-sky conditions, and there never occurs a decrease in DoF. The DoF changes depending on both water vapour and IWP conditions, ranging from 3 (clear, humid conditions) to 10 (very cloudy, humid conditions). The change relative to the preliminary database ranges from 0 to 3. In order to understand this variation in DoF, the regions of Fig. 8 must be considered separately.

In the scenario of high water vapour and low IWP, a DoF of 3 to 4 is found. This is consistent with the three channels that surround each of the water vapour molecule transitions covered. A total of nine ICI channels cover three transitions, leading to redundancy between these channels and thus a relatively low DoF. The majority of this region sees no change in DoF relative to the preliminary database, although a small increase of 1 DoF can be seen for the highest water vapour cases.

Under conditions of low water vapour and low IWP, the DoF increases to 5 and 6 due to increased sensitivity to lower levels of the troposphere and the surface. This corresponds to an increase of 1 to 2 DoF from the preliminary database. This is likely due to the inclusion of V and H polarisation for the 243 GHz channel, allowing for surface effects to be better captured.

For IWP $\geq 40$ g m$^{-2}$ (slightly above ICI's sensitivity threshold), the DoF does not go below 6. A steady increase in DoF occurs with increasing IWP. When comparing to the preliminary database in the region of IWP above the sensitivity threshold, an increase of 1 to 2 DoF is achieved throughout.

The maximum DoF found is 10, occurring in the region of very high IWP and water vapour. This corresponds to an increase of 3 DoF relative to the preliminary database. Although a DoF of 13 would be consistent with the number of channels, such a high value is not necessarily expected due to correlation between the channels (see Fig. 6) and the higher noise present for some channels decreasing their contribution. However, as discussed previously, a relatively large spread can be seen in the joint distributions between all channels, suggesting that every channel contributes to the DoF.

To investigate the contribution of channels to the DoF, we recalculated the DoF across four datasets (results are not shown), each excluding channels centred on a single frequency: 243 GHz, 325 GHz, 448 GHz, or 664 GHz. When excluding the surface-sensitive 243 GHz channels, a significant decrease in DoF occurred for conditions of low water vapour and low IWP, i.e. conditions at which the surface is somewhat visible. The 325 GHz channels and the 448 GHz channels cover water vapour transitions. As expected, removing either of these sets of channels led to a decrease in DoF for all low IWP cases across the entire range of IWP. Exclusion of the 325 GHz channels also lowered DoF for low water vapour and high IWP values, whereas exclusion of the 448 GHz channels lead to a more prominent decrease in the region of high water vapour and low IWP. This suggests that, although both sets of channels cover water vapour transitions, neither set is redundant and both contribute to ICI's overall sensitivity. Finally, removal of the ice hydrometeor-sensitive 664 GHz channel led to a decrease of up to 2 DoF for all high IWP cases. In fact, all channels led to a small decrease of DoF for high IWP values, implying that every ICI channel can play a role in the retrievals.

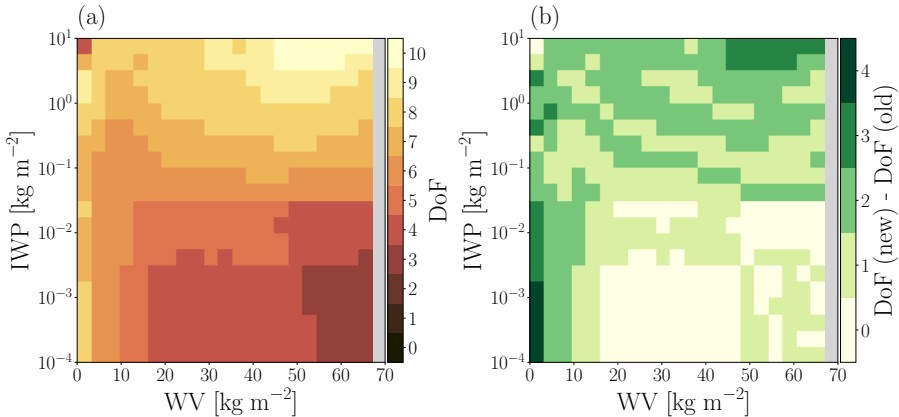

**Figure 8.** The estimated degree of freedom (DoF) of the simulated observations within the ICI retrieval database is presented in panel (a). DoFs are given across total column water vapour (WV) and IWP. Shown in panel (b) is the difference in DoF achieved with the new retrieval database and the preliminary database found in Eriksson et al. (2020). Cases at all latitudes are included.

To assess the impact of the introduction of polarised channels in the database, we conducted an additional analysis of the DoF on a 'polarisation-free' version of the new database (results are not shown). Namely, we replaced the individual observations from V and H polarised channels at the same frequency with their mean value. A decrease of at least 1 DoF was observed over the entire region of $IWP \geq 40$ g m$^{-2}$, with a decrease of 2 in the high-IWP and high-WV region. This leads to a maximum DoF of 8. Despite the decrease in DoF in this region, there still remains an overall positive difference in DoF between the

new database and the preliminary database. In the region of low-IWP and low-WV (conditions under which the surface can be seen), a drop of 1 to 2 DoF occurred, resulting in little difference relative to the preliminary database.

To check the impact of now assuming 75% NE$\Delta$T, another analysis was performed using the original noise values of NE$\Delta$T (results are not shown). The maximum DoF achieved was 8, with an overall decrease in DoF seen across the entire region of high-IWP. Small decreases were seen elsewhere, but these were not major.

Although the analyses performed were relatively simple, they do allow us to infer that both the introduction of polarisation and the reduction in noise have improved the DoF of the database. Additionally, increases to the DoF relative to the preliminary database still occurred when attempting to remove the influence of these factors. We conclude that other developments to the database (see Sect. 3.1) also play a role in the increases of DoF, although further analysis is required to quantify this further.

### 5.3   Ice water path retrievals

A QRNN was trained on a subset of the database and subsequently used to retrieve IWP, $D_{\mathrm{m}}$, and $Z_{\mathrm{m}}$ from observations in a separate subset. The success of the retrievals was evaluated through a comparison of retrieved cases to true cases. Evaluating a probabilistic estimate against a single true value can be challenging unless done on a case-by-case basis. To compare a set of retrieved distributions with the 'truth', a point estimate must be selected from each distribution. There are several options

for choosing a point estimate. One simple option is to calculate the mean of the distribution. To do so, a CDF is constructed by interpolating between the retrieved quantiles and extrapolating to 0 and 1 outside the minimum and maximum quantiles, respectively. The first moment of the predicted PDF provides the mean. An alternative choice of point estimate is to take a single random sample from a posterior distribution, where the posterior distribution is computed through an interpolation of the inverse CDF. By definition, a mean does not contain information on the values present in the tails of a distribution. In contrast, there is some probability that a random sample from the posterior will fall within this region. Therefore, a large set of point estimates obtained by randomly sampling the distributions is expected to capture more extreme cases.

When comparing retrievals directly to the database values, we use the mean as the point estimate. The mean, median, 16% quantile, and 84% quantile of all retrieved means are subsequently calculated as a function of the database values, primarily to provide a visual comparison. The remaining statistics, such as bias and correlation coefficient, are computed on the complete set of posterior means.

Retrieval performance for IWP is presented in panel (a) of Fig. 9. In the case of IWP retrievals, the correlation coefficient is found to be $r = 0.87$. The overall bias for retrievals of IWP $> 1$ g m$^{-2}$ is $-4$ g m$^{-2}$. We note that the bias can be a misleading statistic due to its capacity to be influenced by the density of samples in a particular region, or if the data ranges across orders of magnitude. Alternatively, Fig.9 offers a qualitative, and perhaps more intuitive, representation of the bias. The median follows the true values closely for $10$ g m$^{-2} <$ IWP $< 1.7$ kg m$^{-2}$. Retrievals at IWP $> 1.7$ kg m$^{-2}$ display a negative bias, likely due to fewer of these cases present in the training data. Values in this region are the main contributors to the negative bias. The overall bias is therefore small relative to these influential high-IWP values. IWP data less than $1$ g m$^{-2}$ are not represented in the figure, but have an overall positive bias of $2$ g m$^{-2}$. A positive bias appears in the median at IWP $< 10$ g m$^{-2}$ and in the mean at IWP $< 15$ g m$^{-2}$. The increasing uncertainty and bias of the retrievals in this region can be attributed to the fact that observations become progressively less sensitive to smaller amounts of ice mass, due to the low cloud-impact of these cases. The effect can be seen in the decreasing gradient of the mean and median with decreasing IWP. The overall bias across all IWP retrievals is $0.6$ g m$^{-2}$, indicating that there is a trade-off between the negative and positive bias present for low and high IWP, respectively.

Statistical distributions were made for both retrieved cases and database cases, and subsequently compared to DARDAR (Fig. 10). The comparison is motivated by the fact that successful retrievals are meaningful only if the true cases are themselves realistic. DARDAR IWP was obtained by integrating column-wise all IWC data available in the DARDAR v3.1 product for the year 2010. Both of the aforementioned retrieval point estimates were used and compared to the database simulations and to DARDAR.

Good agreement is seen for all data in the overall distribution (shown in panel (a) of Fig. 10). Since the retrieval distributions match the database, this is an indication that the retrievals are successful in a statistical sense. The model does not appear to be biased within the region of $10^{-3}$ kg m$^{-2} \leq$ IWP $\leq 10^2$ kg m$^{-2}$. This is supported by the model statistics computed for this region (see Fig. 9).

In the zonal mean (shown in panel (b) of Fig. 10), all distributions agree relatively well. Although the retrievals appear higher than the database around the intertropical convergence zone (ITCZ), plotting the zonal mean of the test set (not shown) showed

comparably high values. This indicates that there is no problem with the retrievals in this region; instead, the high values are an artifact of predicting on a subset of all the overall data. At northern mid-latitudes, retrievals appear somewhat low. However, this difference was notably reduced when again plotting the test set (not shown). However, unlike around the ITCZ, retrievals still appeared slightly lower than the test set data at 55°N. We attribute this to poorer retrieval performance at mid-latitudes (discussed further in Section 5.5). Specifically, retrievals at mid-latitudes appear to underestimate the high-IWP cases which have a large influence on the zonal mean calculations.

The database simulations and, by extension, the retrievals achieve a higher probability density of high-IWP values than in DARDAR, which is visible in panel (a) of Fig. 10. The DARDAR distribution drops sharply for IWP $> 10 \, \mathrm{kg} \, \mathrm{m}^{-2}$, whereas the other distributions continue to higher IWP. However, we do not aim to simply reproduce the variability of DARDAR data. In fact, Cazenave et al. (2019) found that changes to the DARDAR-CLOUD product between versions v2 and v3 led to a 24% reduction in IWP on average. It is not automatically true that v3 is 'better' than v2. This emphasises the fact that DARDAR does not necessarily represent the truth, and therefore achieving a perfect match with DARDAR is not essential. Additionally, simulations suggest the possibility of up to $50 \, \mathrm{kg} \, \mathrm{m}^{-2}$ in the presence of tropical deep convection (Bolot et al., 2023). It is therefore possible that DARDAR fails to identify the highest IWP cases. Besides, a greater number of high-IWP cases in the database is not seen as a cause for concern since the ICI retrievals will benefit from more high-IWP cases a priori; such cases are rare and an increase will lead to a better-trained model in this region.

The global distribution of IWP in the database, retrieved IWP, and DARDAR is shown in Fig. 11. The same regional features can be seen in all cases, e.g. ITCZ. The datasets also agree on regions containing fewer ice clouds, such as over the Sahara and Arabian deserts and the stratocumulus regions over subtropical ocean. The retrievals appear noisier, but this is simply attributed to the fact there are fewer retrieval cases than present in the entire database and, to an even greater extent, DARDAR.

## 5.4 $Z_{\mathrm{m}}$ and $D_{\mathrm{m}}$ retrievals

Retrievals of mean mass altitude $Z_{\mathrm{m}}$ have an overall bias of 0.5 km (for cases with IWP $> 10^{-2} \, \mathrm{kg} \, \mathrm{m}^{-2}$) and a correlation coefficient of $r = 0.75$ (panel (b) of Fig. 9). The retrievals correspond well with true (database) values for $Z_{\mathrm{m}}$ between approximately 2 km and 12 km, which is similar behaviour to retrievals performed with the preliminary database (Eriksson et al., 2020). Retrievals of $Z_{\mathrm{m}}$ perform poorer outside this range, though this is as expected.

Retrievals of $D_{\mathrm{m}}$ are presented in panel (c) of Fig. 9. Good agreement is seen for true (database) values below 600 μm, although a slight underestimation is present within the range of 250 μm $< D_{\mathrm{m}} <$ 600 μm. All retrievals corresponding to cases with IWP $> 10^{-2} \, \mathrm{kg} \, \mathrm{m}^{-2}$ contribute to an overall negative bias of $-30$ μm. The overall correlation coefficient is $r = 0.83$. Retrievals of $D_{\mathrm{m}} \geq 600$ μm display a high negative bias, and appear to be increasingly less sensitive to the true value as $D_{\mathrm{m}}$ increases. This is partially due to the fact that higher-$D_{\mathrm{m}}$ values are associated with lower altitudes, at which less information is available from ICI radiances. However, the decrease in sensitivity can also be a saturation effect in terms of ICI wavelengths. Adding the longer-wavelength MWI observations could improve $D_{\mathrm{m}}$ retrievals in this region. At $D_{\mathrm{m}} < 140$ μm, the retrievals display a significant positive bias, with even the 16th percentile of the retrieved mean falling well above the identity line. The retrievals appear to be unsensitive to the true values within this region.

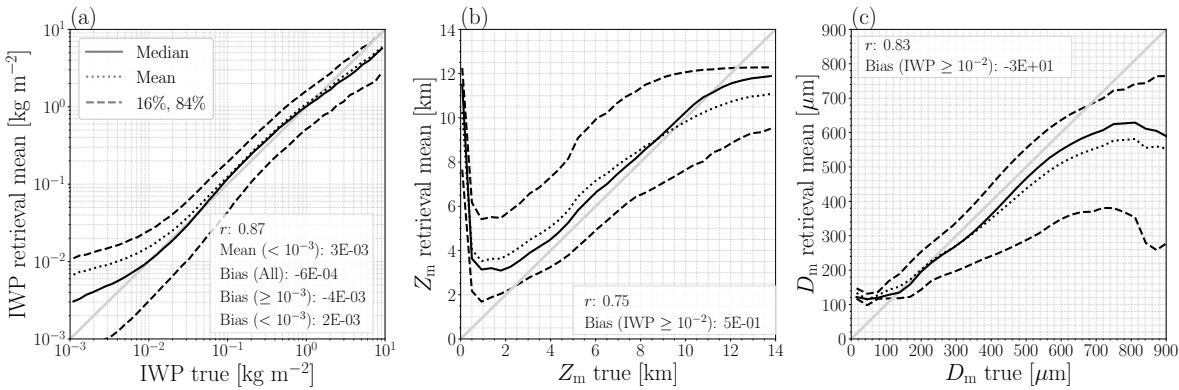

**Figure 9.** Retrieval performance for IWP, $Z_{\mathrm{m}}$, and $D_{\mathrm{m}}$, shown in panels (a), (b), and (c), respectively. The point estimate of the retrieval is taken as the mean of the retrieved posterior distribution. The mean, median, 16th quantile and 84th quantile of the retrieved mean are plotted as a function of the database values (labelled as 'true'). Summary statistics are shown, where $r$ indicates the correlation coefficient. Bias is given in units of kg m$^{-2}$, km, and μm for IWP, $Z_{\mathrm{m}}$ and $D_{\mathrm{m}}$, respectively. Cases of IWP $< 10^{-3}$ kg m$^{-2}$ are not represented graphically, but are included in the relevant summary statistics.

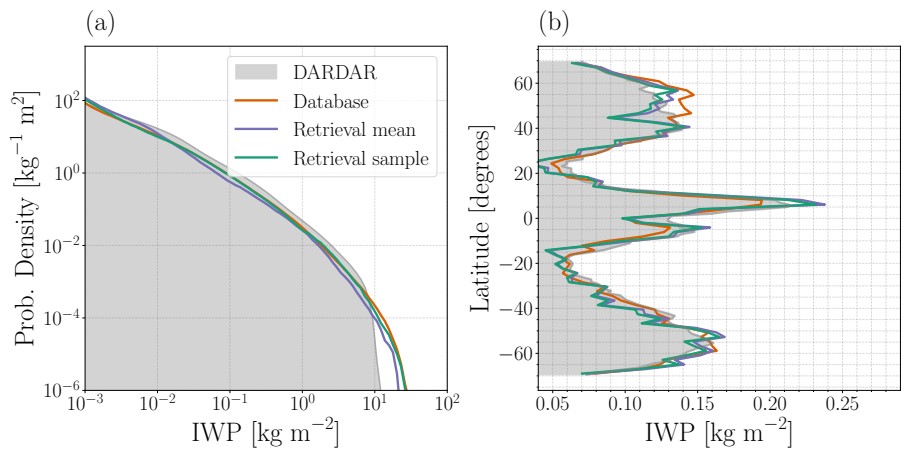

**Figure 10.** Distribution of IWP cases in the retrieval database and retrieved IWP, shown as an overall distribution and as the zonal mean. The retrieval distributions are calculated taking either the posterior mean as a point estimate or by taking a random sample of the posterior as a point estimate. Also shown is the distribution of IWP calculated as the vertical integral of IWC in the DARDAR product for the year 2010.

Overall distributions and zonal means were computed for both database cases and retrieval cases. Results are presented in Fig. 12. In the case of the zonal means, each case is weighted with its corresponding IWP. Although not directly available in the DARDAR product, $Z_{\mathrm{m}}$ and $D_{\mathrm{m}}$ can be calculated by applying Equations 4 and 5 in Eriksson et al. (2020) to variables that are available. $Z_{\mathrm{m}}$ and $D_{\mathrm{m}}$ data corresponding to IWP $< 10^{-2}$ kg m$^{-2}$ were excluded from the distribution calculations, as this data falls below the sensitivity threshold of ICI.

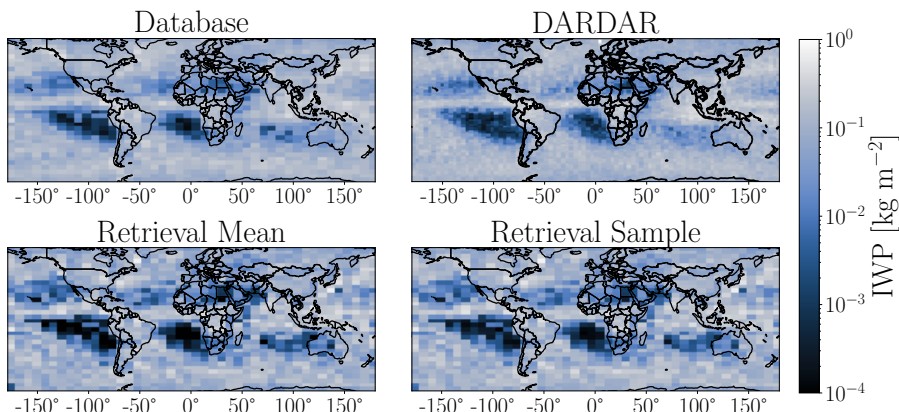

**Figure 11.** The global distribution of IWP cases in the database, retrieved IWP, and IWP calculated from DARDAR from the year 2010. The plot titled 'Retrieval mean' uses the mean of the retrieved posterior distribution as a point estimate. The plot titled 'Retrieval sample' takes a random sample from the posterior distribution as a point estimate.

In panel (a) of Fig. 12, it can be seen that retrievals of $Z_\mathrm{m}$ agree statistically with the database (true values) for $Z_\mathrm{m} < 12$ km, i.e. for all clouds of concern for ICI. As previously discussed, taking the mean as a retrieval point estimate can lead to a
675 failure to capture extreme values in a distribution, as can be seen for $Z_\mathrm{m} > 12$ km in Fig. 12. However, sampling the retrieved posterior distribution does produce a distribution that includes higher $Z_\mathrm{m}$ cases, with such cases appearing with a probability density approximately matching that of DARDAR. This acts as a good example of the benefits of retrieving quantiles, rather than a single estimate.

In the case of the $Z_\mathrm{m}$ zonal means in panel (b) of Fig. 12, database cases, both retrieval estimates, and DARDAR all agree
well. This is expected, since $Z_\mathrm{m}$ is well constrained by the CloudSat reflectivities used to construct the atmospheric scenes. There are some small discrepancies when retrieving $Z_\mathrm{m}$ at very high and low latitudes, but this may be due to difficulties when distinguishing between the surface and low-altitude clouds.

The overall distributions of $D_\mathrm{m}$ are shown in panel (c) of Fig. 12. They are similar in shape and agree somewhat well, but tend to diverge for $D_\mathrm{m} > 700$ μm. This was already evident in Fig. 9 when taking the retrieval mean as a point estimate.
However, even when sampling the retrieved posterior distributions, we fail to capture the extreme values present in the database, indicating that the model struggles to retrieve larger particles in general. The difficulty in retrieving high $D_\mathrm{m}$ is also present in the zonal mean, leading to an overall underestimation.

There is one major difference between our simulations and DARDAR: our simulations contain more higher $D_\mathrm{m}$ cases than DARDAR does. This difference is particularly evident in the zonal mean in panel (d) of Fig. 12. Fortunately, more high $D_\mathrm{m}$
cases probably improves the retrieval model in this region. Furthermore, we again stress that DARDAR is not 'truth', but a retrieval product itself. The discrepancies occur due to differences in particle model assumptions, and serve to demonstrate that we have not tuned our setup to replicate DARDAR. Overall, our $D_\mathrm{m}$ retrievals serve to demonstrate that ICI has sensitivity to

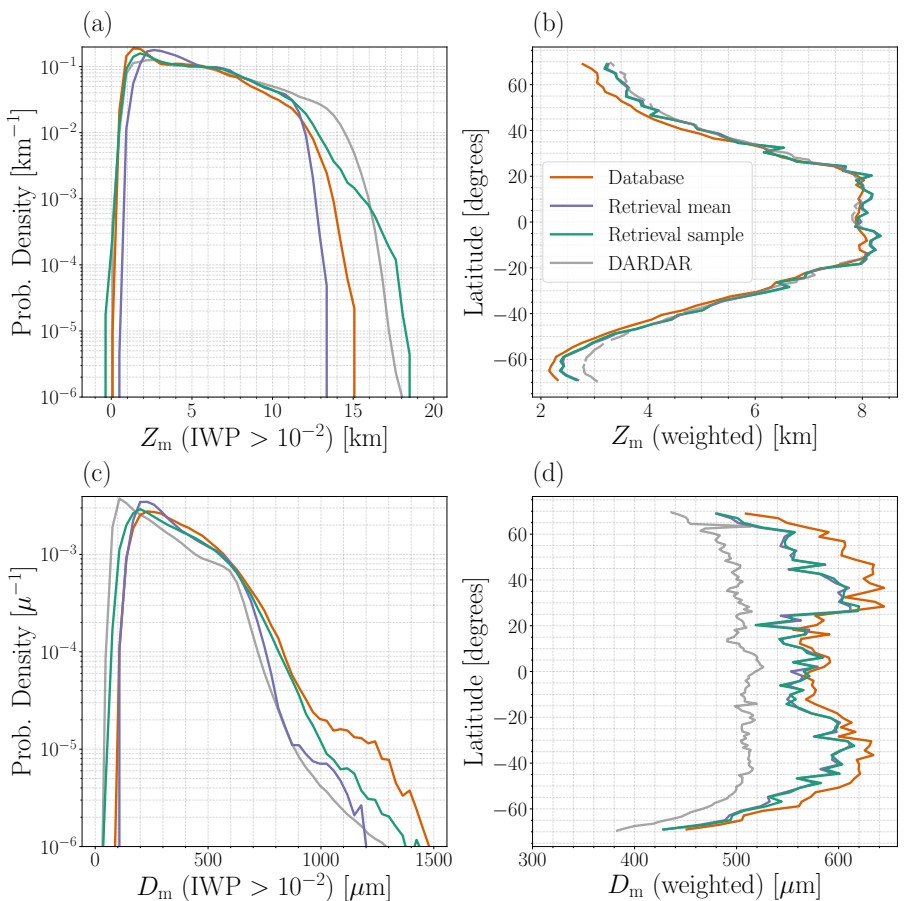

**Figure 12.** Distributions of $Z_m$ and $D_m$, computed from database cases, retrieved means, and random samples from the retrieved posterior. Probability density functions are shown for $Z_m$ and $D_m$ in panels (a) and (c), respectively. Zonal means of $Z_m$ and $D_m$ are presented in panels (b) and (d), respectively. The distributions are calculated using only cases corresponding to IWP $> 10^{-2}$ kg m$^{-2}$, i.e. above the sensitivity threshold for IWP. In the zonal means, $Z_m$ and $D_m$ are also weighted with IWP. Also shown are reference distributions for $Z_m$ and $D_m$ calculated using variables available within the DARDAR product for the year 2010.

$D_m$. This result is supported by Pfreundschuh et al. (2020), which demonstrated that sub-millimetre observations can better constrain microphysical properties.

## 5.5 Retrieval sensitivity

There are multiple factors that may affect retrieval performance. Some effects are related to the sensitivity of the radiances to certain conditions, such as surface type, climatic region, and the optical properties of the ice hydrometeors. For example, it is difficult to isolate surface effects from cloud observations, and therefore surface types associated with higher uncertainties are expected to result in poorer retrievals. Additionally, the latitudinal region will also affect the sensitivity of observations to ice

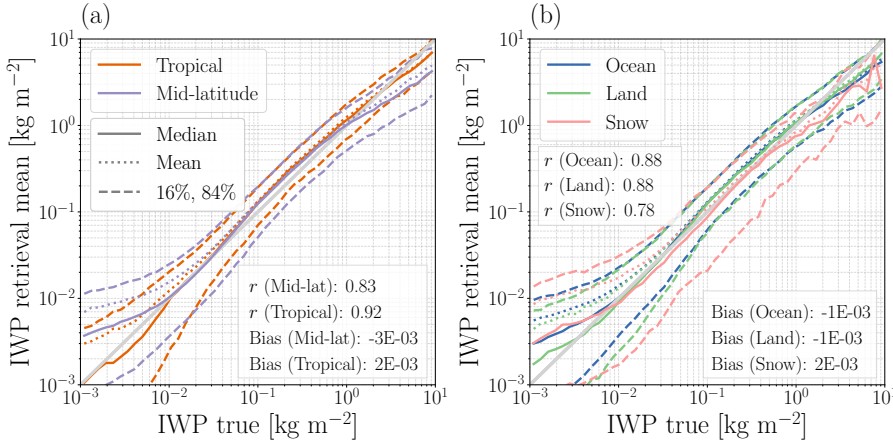

**Figure 13.** Retrieval performance for IWP, separated by latitudinal region in panel (a) and by surface type in panel (b). The mean of the retrieved posterior is taken as the point estimate. Tropical refers to the region bounded by the Tropic of Cancer and the Tropic of Capricorn, i.e. 23.44°N and 23.44°S. Mid-latitude refers to the region bounded by the Arctic circle and the Antarctic circle, i.e. 66°N and 66°S, excluding the tropical region.

mass, due to varying cloud base altitude. Alternatively, retrieval performance can be impacted by the amount of thermal noise present in the observations. This section is an investigation into the sensitivity of retrieval performance to the aforementioned factors. Retrieval performance is evaluated for various latitudinal region and surface type, and across the six particle models used in the simulations. Finally, a simplified assessment of the effect of thermal noise on model training is presented.

Retrieval performance is seen to vary according to latitudinal region. Panel (a) of Fig. 13 demonstrates this accordingly.
Retrievals at tropical latitudes perform better than at mid-latitudes, in the sense that retrievals and mid-latitudes demonstrate a wider spread between the 16% and 84% quantiles of the retrieval mean. Furthermore, the median line follows the truth closely down to as low as $1 \, \mathrm{g \, m^{-2}}$. The finding that retrieval performance improves at tropical latitudes corresponds with results in Eriksson et al. (2020). The decreased performance at mid-latitudes can be attributed to ice clouds lying closer to the surface. To see such clouds, observations must be sensitive to the entire ice column, which is not the case for all channels. However,
channels that are sensitive to the entire column will also produce observations affected by the surface. Surface emissivity effects are still accompanied by significant uncertainties, leading to increasing retrieval uncertainty in the given regions.

The effect of surface type on retrieval performance was also investigated (panel (b) of Fig. 13). Retrieval performance is shown for surface types of ocean, land, and snow. The surface emissivities of snow-covered surfaces remain poorly modelled due to significant variability with frequency. This uncertainty in surface-emissivity, and the fact that this surface type is more
common at higher latitudes, will contribute to a poorer retrieval performance over snow-covered surfaces. At IWP $> 10 \, \mathrm{g \, m^{-2}}$, retrievals of cases over snow-covered surfaces demonstrate an overall negative bias, and a larger spread between the 16% and 84% quantiles than seen for other surface types. The negative bias is more evident with increasing IWP.

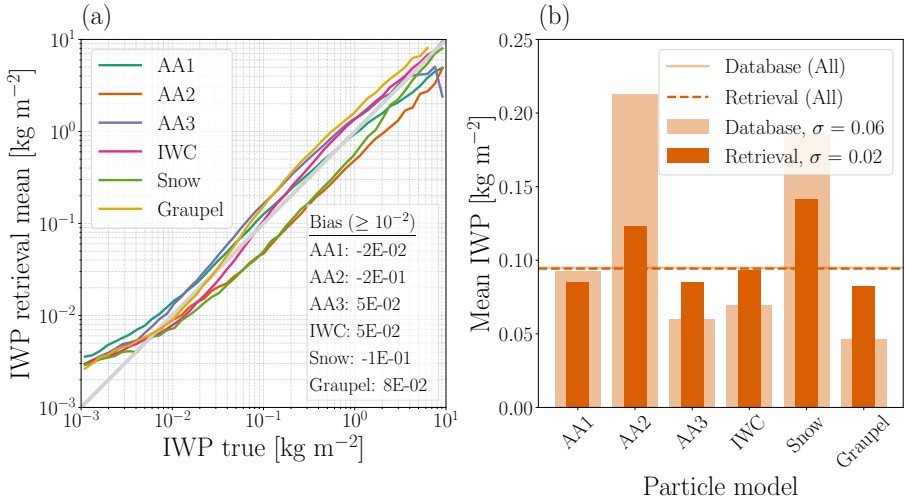

**Figure 14.** Panel (a) displays the retrieval performance for IWP, separated by the particle model used in the simulated case. The mean of the retrieved posterior is taken as the point estimate, and the solid lines correspond to the median of the retrieved means. Bias is computed for all retrievals corresponding to a true IWP $\geq 10^{-2}$ kg m$^{-2}$. In panel (b), the bars shown the mean IWP for a given particle model. Statistics were computed across the test set for both simulated (database) cases and retrieved cases. The solid horizontal line represents the mean IWP for all simulated cases in the test set, and the dashed line represents the mean retrieved IWP for the same cases. The standard deviation $\sigma$ corresponds to the spread of mean IWP between the particle models, and is in units of kg m$^{-2}$.

It is difficult to isolate the effect of surface-type when some surface types are more common in certain climatic regions, e.g. snow at high-latitudes. However, retrievals appear to perform better in tropical regions than at mid-latitudes, despite a less 720 pronounced difference in surface-types between these regions. Due to this, and the physical explanations discussed previously, it is likely that climatic region and surface type independently have an impact on retrieval performance.

The bulk optical properties of the ice particle models implemented in the simulations will have varying effects on the observed radiances. The exact relationship is complex, but the effect of scattering plays a major role. An ice particle with a higher degree of extinction will tend to scatter more radiation away from the sensor's line of sight, thus causing a greater 725 impact on brightness temperatures. For example, Geer et al. (2021) found that the level of extinction exhibited by ARTS particle models approximately correlates to the amount of brightness temperature depression. The dense, rimed hydrometeors used in our study (AA3 and graupel) produce the most scattering and thus have the greatest impact on brightness temperature. Conversely, the low density particle (AA2 and snow) have a weaker effect on brightness temperature.

In the simulations conducted for this study, clouds are assumed to contain only one type of particle. For a cloud with a given 730 IWC, a highly scattering ice particle will cause a greater depression in $T_\mathrm{a}$. Reciprocally, when simulating a cloud composed of highly scattering ice particles, less IWC is needed to replicate the same $T_\mathrm{a}$ when compared to a cloud consisting of particles with weaker scattering.

In an ideal scenario, the retrieval model has the capability to infer the particle model used in a simulation and adjust its IWP prediction accordingly. Alternatively, our model may assume a 'mean particle model' that exhibits the average level of extinction observed across all six ice particle models. If this is the case, then the accuracy of the retrievals can be affected. For instance, if a dense, highly-scattering particle was used in a simulation, the model may overestimate IWP in an attempt to align its prediction with the simulated $T_a$. The converse will also be true; If a particle with lower relative extinction was used in a simulation, the model may underestimate IWP.

To investigate the sensitivity of the retrievals to the choice of particle model, we computed retrieval statistics for each particle models individually (panel (a) of Fig. 14). Above the sensitivity threshold of IWP $= 10^{-2}$ kg m$^{-2}$, there is a trend of overestimation in the case of AA3 and graupel. Likewise, IWP retrievals are generally underestimated in the case of AA2 and snow. Based on these findings, we conclude that the retrievals show sensitivity to the particle model used in the simulations. This suggests that the model generally assumes some average level of extinction. In panel (b) of Fig. 14, we present the mean IWP associated with each particle model for the same set of data, for both simulated and retrieved cases. The spread of bar heights indicates variability between the particle models within the simulated (pre-retrieval) data ($\sigma_{\mathrm{pre}} = 0.06$ kg m$^{-2}$). This variability decreases post-retrieval ($\sigma_{\mathrm{post}} = 0.02$ kg m$^{-2}$). However, the variability does not disappear completely. In conclusion, it may still be possible that the model can distinguish between particle models, but cannot do so with total reliability. An in-depth analysis would be required for further insights. An interesting avenue for such an analysis would involve attempting to retrieve the particle model itself.

Noise is incorporated into the retrievals at two stages; Noise was added to the observations during model training, and to the observations in the test dataset prior to inverting them. To investigate the impact of NE$\Delta$T, two tests were made that alter the inclusion of noise (results not shown). In the first test, the original model, trained with noise, was used to invert the cases in a noise-free test dataset. In the second test, the model was re-trained without noise added to the training observations. Next, noise-free observations were inverted. The second test is equivalent to assuming NE$\Delta$T$= 0$, i.e. an ideal but unrealistic observation with no thermal noise. Only a very small increase of $r$ was seen in the first case, with $r = 0.88$ for IWP retrievals. In the second case, a decrease to $r = 0.79$ was observed for IWP retrievals. This is likely due to the model overfitting to the data in the total absence of noise. In both cases, only very minor visual differences were seen in the equivalents of Fig. 9. We conclude that the current values of NE$\Delta$T have little impact on the training of our retrieval model. This implies that the retrieval performance discussed in this study can be attributed to the quality of the database simulations and to the sensitivity of ICI, rather than the influence of noise.

## 5.6 Limitations

Although efforts were made to produce the most realistic ICI simulations available, there remain limitations. Firstly, although a range of particle models are used, melting ice particles are not considered. Single scattering data for mixed phase particles is not currently available for sub-millimetre frequencies, and calculating such properties would be computationally costly (Kanngiesser and Eriksson, 2022). A second limitation arises from the simplification of assuming the same particle model across an entire column or footprint. This is an unphysical assumption (Kim et al., 2024) that likely affects the realism of the

simulations. Another limitation relates to the challenge of representing three-dimensional variability in the simulations. To this end, the Barker method was applied. However, it is possible that this method underestimates the true variability.

It is difficult to identify all weaknesses in the absence of real, global ICI data. After the launch of ICI, there will be the possibility to compare to real observations. A comparison of observations and simulations at 243 GHz will allow a better assessment of the surface emissivity parameterisation. On the other hand, a clear-sky comparison using channels with lower surface sensitivity will provide insight into the gas absorption models. It will be more complex to identify limitations for cloud-impacted cases, but an investigation of polarisation differences could provide insight into the success of the aARO scheme. It is noted that, in only ten days, ICI will generate the same number of observations as in the retrieval database. Statistically, there will be observations that lie outside the variability of the simulations. However, if it is found that the simulations clearly do not cover the variability of real ICI observations, investigation will be required.

## 6 Summary and conclusions

In this study, we presented the state-of-the-art retrieval database developed for use within the operational ICI retrieval algorithm. The database generation strategy was based upon Eriksson et al. (2020), but multiple improvements were made. Among the most significant extensions were the inclusion of the full channel responses and representation of the full two-dimensional antenna pattern. Additionally, the treatment of ice hydrometeors now incorporates an approximate treatment of particle orientation, allowing for polarisation effects to be captured. Even taking into account the approaches taken to improve calculation efficiency, the simulations performed within this study are, to our knowledge, the most detailed performed to date.

In the first part of the study, we performed an analysis of simulated antenna temperatures. The degrees of freedom of the antenna temperatures were computed and results were compared to the same method applied to the preliminary database of Eriksson et al. (2020). An overall increase in the degrees of freedom was seen, indicating that simulations are more realistic and that improvements made to the database have increased the information content of the simulated radiances.

We also simulated radiances as observed by existing instruments, in order to validate the simulation methods while bypassing the need for real ICI observations. The database generation framework was used to simulate antenna temperatures as would be observed by sensors on the flight campaigns ISMAR and MARSS. This allowed for a comparison of both microwave and sub-millimetre observations. The simulations were validated against real observations and were found to cover the entire measurement space. Likewise, we simulated four high-frequency microwave GMI channels. The simulated radiances overall followed the same statistical distribution as real GMI observations, and covered the measurement space in most cases. Some differences were observed, but occurred only for more extreme brightness temperatures, and can partially be explained by too few simulated cases.

In the second part of the study, we characterised the expected retrieval performance associated with using the database within an inversion scheme. A probabilistic machine learning model was trained on the database simulations. The model then performed inversions of simulated antenna temperatures to retrieve ice water path, mean mass height, and mean mass diameter. The results indicate a sensitivity to IWP between $10 \, \mathrm{g \, m^{-2}}$ and $10 \, \mathrm{kg \, m^{-2}}$. The success of retrievals at the higher end of the

sensitivity range and above is constrained by the number of similar cases used to train the model, and such cases seldom occur in reality. This highlights the importance of representing these cases in the database.

Additionally, we show that both simulated and retrieved IWP and $Z_\mathrm{m}$ data are statistically consistent with the DARDAR product. Simulated and retrieved $D_\mathrm{m}$ cases slightly differ from DARDAR in their distributions, demonstrating that our retrievals are not simply a replica of another product.

In this study, we decided to retrieve only the variables that will be offered in the EUMETSAT L2 product. However, it is possible that the radiances contain enough information to retrieve other variables. For example, due to the high number of ICI channels, observations could also provide information on the vertical structure of ice hydrometeors. This possibility warrants further investigation, since it would expand the already powerful dataset offered by ICI.

Eriksson et al. (2020) estimated that ICI will deliver similar accuracy to the CloudSat- and CALIPSO-based DARDAR product within the sensitivity range of IWP, and we keep this opinion. Good retrieval accuracy was maintained for $Z_\mathrm{m}$ and $D_\mathrm{m}$, even with the more diverse set of particle models used. In regard to $Z_\mathrm{m}$, ICI cannot compete with radars. However, we have shown that it is possible to retrieve $D_\mathrm{m}$ from ICI observations. Given that ICI spans microwave and sub-millimetre wavelengths with multiple channels, it is possible that there is more information available to constrain $D_\mathrm{m}$, allowing ICI to contend with radars.

Notably, both the CloudSat and CALIPSO satellites ended operations in 2023. Since alternative data sources on atmospheric ice mass suffer from inconsistencies and uncertainties, we are now left with a significant observational gap. Yet, the future is promising; EarthCare (the Earth Cloud Aerosol and Radiation Explorer) was recently launched in 2024 (Illingworth et al., 2015) and ICI will be launched in the following years. EarthCare offers radar-lidar cloud profile measurements, acting as a continuation of the CloudSat and CALIPSO pairing. As this study has demonstrated, ICI will offer observations specially suited to the measurement of ice hydrometeors. With these new sources of data, we move closer to achieving the reliable and consistent global datasets of ice mass we need.

### Appendix A:  Machine learning

Machine learning problems are typically used to predict an output $\boldsymbol{y}$ from a given input $\boldsymbol{x}$ such that $f(\boldsymbol{x}) = \boldsymbol{y}$. The model $f$ learns to map input to output from a training set $\{\boldsymbol{x}_i, \boldsymbol{y}_i\}_i^n$.

A QRNN is a machine learning approach designed to provide an estimation of the quantiles of the cumulative distribution function of a prediction output value. The $\tau$th quantile $x_\tau$, where $\tau \in [0,1]$, of a cumulative distribution function $F(x)$ can be defined as follows (Pfreundschuh et al., 2018):

$$x_\tau = \inf\{x : F(x) \geq \tau\}. \tag{A1}$$

A QRNN is essentially an extension of machine learning regression models trained to predict the median. To this end, a QRNN employs a quantile loss function that calculates the loss between predicted and target values for a given quantile level $\tau$. This

allows the variability of the prediction to be captured. The quantile loss function is defined as:

$$\mathcal{L}_\tau(x_\tau, x) = \begin{cases} \tau|x - x_\tau|, & x_\tau < x \\ (1 - \tau)|x_\tau - x|, & \text{otherwise.} \end{cases} \tag{A2}$$

The expectation of $\mathcal{L}_\tau(x_\tau, x)$ with respect to $x$ is minimised by $x_\tau$. Therefore, given the training data $\{\boldsymbol{x}_i, \boldsymbol{y}_i\}_i^n$, the model $f$ is trained to minimise the expectation of $\mathcal{L}_\tau(f(\boldsymbol{x}), \boldsymbol{y})$. Extending the model to predict multiple independent quantiles allows

a discrete approximation of the cumulative distribution function (CDF) $F_{y|\boldsymbol{x}}(y)$ to be made. From a Bayesian perspective, the PDF derived from the CDF can be thought of as a posterior distribution that incorporates the a priori knowledge inherent in the training data. We note that machine learning terminology is used here; $\boldsymbol{x}$ is generally used in machine learning to represent the input to the model, which in this case is a measurement. The same measurement is typically represented as $\boldsymbol{y}$ when used in the context of the forward model and the inverse problem.

A single model is trained to handle all observations in the retrieval database, i.e. all surface-types and latitudinal regions are included. The input vector $\boldsymbol{x}$ includes observations from the 13 ICI channels, in the form of antenna temperatures $T_\mathrm{a}$. The use of $\Delta T_\mathrm{a}$ as a model input (as to be used in the L2 product) was also checked, and no major difference in retrieval performance was observed between the two variables (Appendix C). Surface temperature, surface pressure and surface classification are also included within the input as ancillary data. The output of the model are 99 quantiles of the CDF $F_{y|\boldsymbol{x}}(y)$ for each of the

variables IWP, $Z_\mathrm{m}$ and $D_\mathrm{m}$.

## A1    Neural network architecture and input data

The architecture of the QRNN used to perform the retrievals in this study is as follows:

- A single network was trained to predict all outputs IWP, IWC, $Z_\mathrm{m}$, $D_\mathrm{m}$. The network consisted of multiple fully-connected layers for all outputs, with several final layers for each output individually.

- The QRNN was trained to predict 99 uniformly spaced quantile levels $\tau \in [0.01, 0.99]$ for all output variables.

- Hyper-parameters, such as number of neurons, number of layers, and batch size were selected after training multiple networks and subsequently comparing their performance on the test dataset.

The training and validation data was structured and processed as follows:

- The input data consisted of antenna-weighted brightness temperatures for each of the 13 ICI channels. A classification
of surface type, surface temperature, and surface pressure (included as variables in the retrieval database) were included as ancillary input data.

- Uncertainty due to thermal noise is generated as a Gaussian with zero mean and variance $\mathrm{NE}\Delta\mathrm{T}_j^2$, where $j$ is the ICI channel. To avoid overfitting, noise was randomly generated and added to each batch of training data, during each epoch of training. Also in each training epoch and batch of training data, 1% of the surface types were shuffled, both to avoid
overfitting of the model and to safeguard against any misclassification of surface type in the dataset.

**Table B1.** Definitions of the MARSS and ISMAR channels simulated in this study. Not all channel IDs are official, and are used only in this study to simplify their representation.

| Sensor | Channel Frequency (GHz) | Polarisation | Channel ID |
|--------|------------------------|--------------|------------|
| MARSS | $157.07 \pm 2.60$ | H | |
| MARSS | $183.25 \pm 3.00$ | H | |
| MARSS | $183.25 \pm 7.00$ | H | |
| ISMAR | $243.20 \pm 2.50$ | H | |
| ISMAR | $243.20 \pm 2.50$ | V | |
| ISMAR | $325.15 \pm 1.50$ | V | |
| ISMAR | $325.15 \pm 3.50$ | V | |
| ISMAR | $325.15 \pm 9.50$ | V | |
| ISMAR | $664.00 \pm 4.20$ | H | |
| ISMAR | $664.00 \pm 4.20$ | V | |

– Due to the large range of magnitudes possible for IWP, a log-linear transformation was applied, where all values of IWP less than $1.0 \ \mathrm{kg \ m^{-2}}$ were transformed into logarithmic space. Additionally, any cases of IWP less than $10^{-4} \ \mathrm{kg}$ $\mathrm{m^{-2}}$ were replaced with a value uniformly sampled from the range $[10^{-6}, 10^{-4}]$ and subsequently transformed in to logarithmic space. The remainder of the data were linearly normalised according to the statistics of the training set.

## Appendix B:  ISMAR and MARSS channels

Rather than refer to each ISMAR and MARSS channel by its specific frequency in Sect. 5.1.2, IDs were assigned to each channel. The channel IDs are presented in Table B1. The numbering of the MARSS channel IDs corresponds to the official IDs (McGrath and Hewison, 2001). The channel IDs chosen for ISMAR channels are not official. The numbering of the ISMAR channels simply follows the order in which they appear in the table, with the same number applied to channels at the same frequency and bandwidth.

## Appendix C:  Cloud signal-based retrievals

Retrievals were performed using a model trained with simulated ICI $T_\mathrm{a}$ as the primary input. In contrast, the operational L2 product at EUMETSAT will perform inversions of $\Delta T_\mathrm{a}$. Our decision to focus on $T_\mathrm{a}$ is driven by the desire to minimise the complexity of the retrieval problem.

However, given that $\Delta T_\mathrm{a}$ is a key component in the retrieval algorithm, we wished to test the use of $\Delta T_\mathrm{a}$ as a model input in order to check that there was no major disadvantage arising from the choice to use $T_\mathrm{a}$. We continued to exclude the pre-

processing steps performed in the algorithm, thus performing a direct comparison between $T_a$ and $\Delta T_a$ when used as retrieval model inputs.

A QRNN was trained on the same training and validation sets as for the original model. The model was trained using the same ancillary inputs (see Appendix A1), but the primary input changed to $\Delta T_a$ values for each ICI channel. Inversions were performed on the same test data partition as originally used. In other words, $\Delta T_a$ are inverted to obtain IWP, $Z_m$, and $D_m$, where the 'true' values of IWP, $Z_m$, and $D_m$ are the same for both retrieval models.

Both models exhibited very similar behaviour when predicting all three variables (Fig. C1). The most obvious differences occur at the extreme ends of the shown ranges. Both models produce the same correlation coefficient (at the precision shown), and this behaviour is again seen across all three variables. In the case of IWP, a bias of $-6 \times 10^{-4}$ kg m$^{-2}$ is seen for the $\Delta T_a$ model. This is marginally smaller than the bias of $2 \times 10^{-3}$ kg m$^{-2}$ of the $T_a$ model, and notably negative rather than positive. The bias of the $D_m$ $\Delta T_a$ predictions is slightly higher than for the $T_a$ model, at 3 μm as opposed to 2 μm.

The consistency in performance between the two models suggests that the all-sky simulations $T_a$ appear equally capable as $\Delta T_a$ in the context of machine learning retrievals. However, it is stressed that this conclusion is made in the context of machine learning retrievals, and cannot necessarily be extended to the BMCI method used for the L2 product.

Explaining the minor discrepancies between the models is somewhat challenging. The bias was computed across the entire test dataset and not limited to the dataset represented in Fig. C1. It is therefore likely that the difference in bias occurs in regions where the model is less well-trained. This is supported by the fact that the inversions produce almost identical results in regions where we expect the model to be well-trained, such as above the sensitivity threshold of ICI and where there exists a large amount of data (e.g. $10^{-2}$ kg m$^{-2} \leq$ IWP $\leq 1$ kg m$^{-2}$). Given the small scale of the differences, the differences could be attributed to the nature of neural networks. Although the two networks were similarly configured, the learning process and final weights will not be identical, thus accounting for the small variation between models.

*Code and data availability.* The code used for analysis and plotting is available at https://doi.org/10.5281/zenodo.10839090 (May, 2024). The data pertaining to ICI used in this study are available under license for non-commercial purposes and on the condition of no redistribution by contacting EUMETSAT (vinia.mattioli@eumetsat.int). DARDAR-cloud v3-10 data are available at https://www.icare.univ-lille.fr/dardar/data-access/ (last access: 19 March 2024). GPM GMI data were accessed via NASA's Goddard Earth Services Data and Information Services Center (GES DISC), at https://gpm.nasa.gov/ (last access: 19 March 2024).
ISMAR and MARSS observations from Facility for Airborne Atmospheric Measurements (FAAM) flights were accessed via the NERC CEDA archive. The specific flights used were (with revisions indicated by "R" and flight numbers by "B"): R014_B893, R014_B895, R014_B896, R014_B897, R014_B898, R005_B939, R005_B940, R005_B941, R005_B945, R005_B949, R006_B951, R006_B952, R001_B984, R002_C153, R002_C156, R002_C157, R002_C158, R002_C159, R002_C160, R002_C161, R002_C164, and R002_C168, which can be found at Facility for Airborne Atmospheric Measurements et al. (2016a; 2016b; 2015; 2016c; 2016d; 2016e; 2016f; 2016g; 2016h; 2016i; 2016j; 2016k; 2016l; 2019a; 2019b; 2019c; 2019d; 2019e; 2019f; 2019g; 2019h; 2019i).

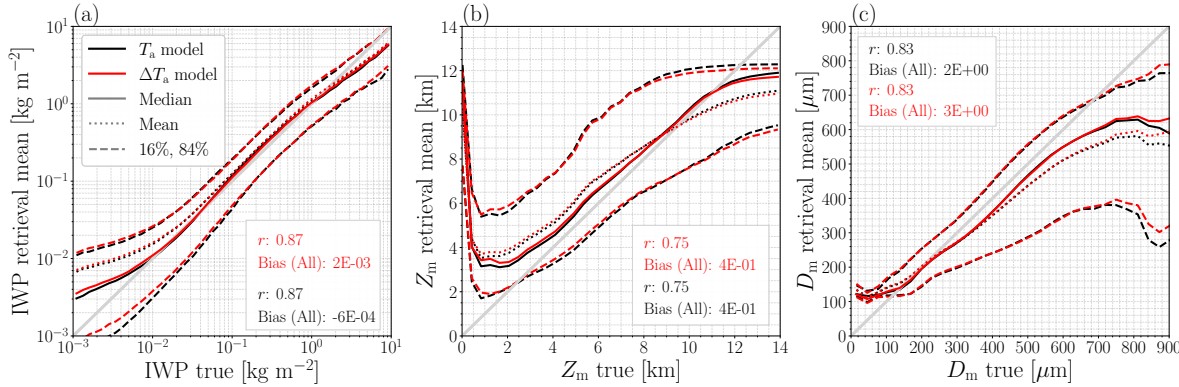

**Figure C1.** Retrieval performance for IWP, $Z_\mathrm{m}$ and $D_\mathrm{m}$. The predictions from the $T_\mathrm{a}$ model, shown in black, are the same as shown in Fig. 9. The predictions from the $\Delta T_\mathrm{a}$ model, shown in red, are a result of training a model to predict given a vector of $\Delta T_\mathrm{a}$, instead of $T_\mathrm{a}$. The ancillary input data and outputs were kept the same in both models. $r$ indicates the correlation coefficient. Bias is given in units of kg m$^{-2}$, km, and µm for IWP, $Z_\mathrm{m}$ and $D_\mathrm{m}$, respectively.

*Author contributions.* All authors contributed to the development of the database generation framework. PE provided input on all results and contributed to revisions of the paper. EM led the writing of the manuscript, and performed the visualisation and analysis of the results, with the exception of the GMI simulations which were visualised and analysed by HH. BE performed the ISMAR simulations and HH performed the GMI simulations.

*Competing interests.* The authors declare that they have no conflict of interest.

*Acknowledgements.* The development of the retrieval database was performed as part of the EUMETSAT study "Development of a cloud radiation database for EPS-SG ICI". Retrievals were performed using the QRNN package developed and made available by Simon Pfreundschuh, available at https://github.com/simonpf/quantnn, (last access: 22 February 2024).

The authors would like to thank the crew and personnel who were involved in the COSMICS, CIRCCREX, WINTEX, ISMAR, T-NAWDEX, and PIKNMIX-F flight campaigns, which provided the ISMAR and MARSS observations used in this study. The BAE-146 aircraft is operated by Airtask and Avalon and managed by the Facility for Airborne Atmospheric Measurements (FAAM), funded by the Natural Environment Research Council (NERC) and the Met Office.

The authors would also like to thank both Simon Pfreundschuh and Adrià Amell for their input and guidance on machine learning.

Computations were performed using resources at Chalmers Centre for Computational Science and Engineering (C3SE) provided by the Swedish National Infrastructure for Computing (SNIC).

*Financial support.* The research was partly funded by the Swedish National Space agency (grant no. 2021-00077). Development of the database was performed under the EUMETSAT study "Development of a cloud radiation database for EPS-SG ICI", contract EUM/-CO/21/4600002601/VM.

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
