# Peer review of "The Ice Cloud Imager: retrieval of frozen water column properties"

_EGUsphere, 2024_

## Author Comment (AC1)

**Response to comments from Anonymous Referee #1**

July 17, 2024

We thank the reviewer for taking the time to read the manuscript and provide detailed and valuable feedback.

**Specific comments**

Line 24: I would leave out "now" in this sentence, as the reference is 30 years old
Fixed.

Line 44: the authors list the MWI frequencies later in the paper, but a range (19-183) would be useful here since it specifically mentions the frequency coverage.
To avoid repeating the range (since it is mentioned later), the phrase has been edited to now say: "extending the coverage down to 18.7 GHz".

Line 72: is the overpass time known? 1:00?
The local time of the descending node is 9:30, and this information has now been included when describing ICI.

Line 89: reword "precipitation and snowfall" as both are precipitation. Suggest "liquid and frozen precipitation" or "rainfall and snowfall"
The phrase has been updated to say "liquid and frozen precipitation".

Line 170: this is done in section 3.2, but I would suggest adding information right at the beginning of this discussion about the period and coverage used, as I was asking this while reading this section. (two years, global CloudSat overpasses) and ancillary model data used
Information about the coverage, period, and the use of ERA5 ancillary data has now been included as the second paragraph in Section 3.1, since the choices were motivated by the requirements on the database listed in the previous paragraph.

Figure 2: bottom panel - would prefer frequencies rather than channel#. Should the y axis be labeled Ta rather than Tb?

Yes, the y axis should say Ta. This is fixed, and the legend and figure caption have been updated to give frequencies rather than channel number.

Line 451: suggest add "conically scanning"
Added.

Line 461-465: Just a note that this is almost certainly the case and emphasizes the need for better understanding of emissivity at these higher frequencies
This is a good point, and certainly a limitation of such simulations. The need for better understanding of emissivities is now explicitly stated in this discussion.

Figure 5-6: would prefer labeling with center frequency rather than channel number (lots of flipping between pages)
Both figures 5 and 6 have been updated to state the channel frequency and the channel polarisation (when necessary).

Line is the same 614: Not sure if I missed it but should have a brief introduction of DARDAR somewhere
A description of DARDAR and the satellites that the product uses is now given when DARDAR data is first discussed (at the end of Section 3.3).

Line 618: There are a couple of features in the zonal mean plot that I think are worth discussing. Retrieved IWP is lower than the database for the northern midlatitudes. Perhaps a land/vegetation issue? Retrievals are conversely high for the ITCZ. Thoughts on this?
This is a good point, and we have now looked into this further. Plotting a zonal mean of the test set, i.e. the 'true' IWP corresponding to each retrieved case, leads to equally high IWP around the ITCZ. This implies that there is not a problem with the retrievals in this region, since it is present in the data too.
Approximately 10,000 scenes were randomly selected to form the test set. This is around 20% of the total number of scenes. This is not enough to perfectly reproduce the zonal mean of the full database at every latitude. However, good agreement is seen overall.
In regards to the northern mid-latitudes, the conclusion was similar. The zonal mean of the test set shows lower IWP at such latitudes, agreeing more with the retrievals. Small discrepencies still remained, but they were not as significant as seen in the comparison of the retrievals and the full database. We believe that this is likely due to generally poorer retrieval performance at mid-latitudes, due to both lower altitude clouds and potentially to increasing uncertainty in surface emissivities as snow cover increases. Since this is important to note, the above findings are now included. The test set zonal mean has not been included in the original figure to avoid the plot becoming too confusing. Instead, the findings are described in the text. Discussion of the zonal mean has also been moved into a seperate paragraph to improve the clarity.

Line 784: Somewhere here it seems worth mentioning the importance of rep-

This is true, particularly for high cases. The importance of including high values in the database to achieve retrievals of such cases is now emphasised in the conclusion. The lower cases are influenced by more than just their representation in the database. Although retrievals of low values appear somewhat uncertain, there are in fact many of these cases in the database. Instead, retrievals of the lower values are more influenced by the sensor sensitivity. Furthermore, particularly low cases (less than 0.1g/m2) are not retrieved. Therefore, we avoid placing emphasis on only the importance of representation in this case.

---

## Author Comment (AC2)

**Response to comments from Anonymous Referee #2**

July 17, 2024

We thank the reviewer for taking the time to read the manuscript and provide detailed and valuable feedback.

**General comments**

- There are inconsistencies in the formatting of the text preceding equations in the report. For example, a colon (:) is used before equations in some instances (e.g., line 374), while in others, no punctuation is used.
  Fixed.

- The most precise way to refer to Metop-SG, is MetOp-SG.
  Fixed.

- When a citation is given in parentheses, remove redundant brackets, e.g. (see Table 1 in Eriksson et. al (2020)) =¿ (see Table 1 in Eriksson et. al, 2020).
  Fixed.

- When a panel of a figure is described, please include the figure number as well for increasing clarity.
  Fixed.

- When the discussion is centred around results that are not presented within the manuscript, please add the relevant info (in the beginning of the paragraph), e.g. not shown here.
  Fixed.

- On some occasions, a space is missing between a number and its unit.
  Fixed.

- Section 4.1, could be moved to the Appendix
  It is agreed that this would be more suitable. Fixed.

- Please consider providing information on DARDAR and CALIPSO. Initially, the acronyms are not defined. Furthermore, enhancing clarity by

including relevant details about the type and usage of any datasets referenced would greatly benefit readers.

**Fixed the acronyms and DARDAR/CALIPSO information. Further information regarding exactly what geophysical data (i.e. ECMWF data) that we include in the simulations has been added in the text. The reference to more details on such data has been made clearer, explaining where the reader may find out more about the humidity profile.**

**Specific comments**

The following acronyms have not been introduced:

- EUMETSAT (mentioned in Affiliations, Abstract, and the first reference in the Introduction)

- DARDAR (line 15)

- ARTS (line 135)

- RTTOV (line 136)

- ECMWF (line 145)

- DISORT (line 304)

- CALIPSO (line 791)

- CloudSat (line 158)

- EarthCare (line 799)

**All acronyms, except CloudSat, have now been introduced in the manuscript. A reference to CloudSat has now been included directly after its first use.**

On the other hand, the QRNN acronym (line 418) has already been defined.
**Fixed.**

Line 24: remove "now"; reference is more than 30 years old. The authors could consider updating the grammar as well.
**Fixed, and the sentence is updated to improve flow.**

Line 37: Adding some examples or references to passive optical and infrared missions that capture cloud top data could be beneficial.
**MODIS and SEVERI are now included as examples of such missions.**

Line 43: I have never seen the acronym to be written as Micro-Wave. Why not use the most common mention, i.e. Microwave Imager?
**Fixed.**

Line 44: The authors could consider including the MWI frequency range.

The comment "extending the coverage down to 18.7 GHz" has been added when introducing MWI for the first time. The full frequency range is also given when describing MWI in more detail in Section 2.1.

Line 71: Since the authors decided to add information on MetOp-SG A, including details on the differences between the two missions could be valuable. To elaborate, MetOp-SG A focuses on optical/infrared missions and atmospheric sounders, while MetOp-SG B focuses on microwave instruments. Otherwise, the mention does not provide any benefit. Please consider including any citation.

The above information has been included when introducing the pair of satellites. To avoid giving too much detail here, a citation is given for readers who wish for more information.

Line 104: There is no information on what an L1b product is. Plus, consider defining the acronym Level 1b (L1b).

Added the acronym definition and defined L1b in the context of ICI as "calibrated and geo-located antenna temperatures".

Table 2: A full stop is missing at the end of the caption.

Fixed.

Line 89-90: precipitation and snowfall are both precipitation; maybe reward it to "liquid and solid precipitation".

This has been reworded to "liquid and frozen precipitation".

Line 94: "using different methods"; the word "retrieval" is redundant due to the aforementioned.

The word retrieval has been removed.

Line 235: No subscript should be in italic; please check this throughout the paper.

Fixed in both the text and all plots.

Line 251: There is no information on what the DARDAR product is. A brief introduction, since it is being used, could add value to the paper.

When introducing DARDAR, it is now specified that it is a product offering IWC retrievals. Furthermore, the two satellites (CloudSat/CALIPSO) that DARDAR is based on are introduced.

Line 260: "The scaling differs between V and H polarisation", Could the authors elaborate on this a bit further? Please have a look at Barlakas, Geer, and Eriksson 2022 (Cloud particle orientation and polarisation for cross-track microwave sensors; NWP-SAF).

It is agreed that the method and reasoning for differing the scaling between polarisations is not well justified in the manuscript. The text has now been edited

to explain that the extinction is scaled differently for V- and H- polarisation such that the polarisation difference increases. Since the range of aARO factor used is mentioned later, it was also thought useful to explain what the lower limit corresponds to, i.e. an aARO factor of 1.0 corresponds to TRO. Furthermore, the above reference is now used to explain that larger scaling is required for H-polarisation.

Line 304: To begin with, DISORT acronym has not been introduced. Why do not you add some brief information on what the DISORT solver is, including a reference.
A short description of DISORT, including the method used, has now been included, alongside a citation.

Line 466: This can imply that the particle models could be further tailored. Could you please add some comments?
This is an interesting comment, and makes a good point. Although there is good agreement with GMI, even better agreement could be improved by adjusting the particle models. However, the authors believe that it is best to wait to until real ICI data is available to do so. Upon comparison with real ICI data, there will certainly be several areas in which improvements could be made. Therefore, it is best to avoid tailoring too much to match other instruments now and risk re-tailoring later. However, the manuscript did not previously state the plan to compare and tailor to real ICI observations later. This has now been added to the text in Section 5.1.1, explaining why we do not tailor to GMI and instead will tailor to ICI.

Figure 3: a space is missing from [-60, 60].
Fixed.

Line 519: Any comments on the differences in the polarisation signature between 243 and 664 GHz? Excluding the surface contamination, at which frequency do you expect to see larger polarisation differences and why (considering the ice amount)?
Thank you for this interesting comment. The amount of polarisation difference will depend on several factors, including frequency, particle orientation, particle habit, and particle size. For example, smaller particles will be less likely to be oriented, and the higher frequency channels (664 GHz) are most sensitive to such particles. This would likely lead to lower polarisation differences at 664 GHz. There will likely also be an interplay between the above factors and altitude.
Such effects may be visible in real observations, if it were possible to exclude surface effects. However, the ICI antenna temperatures shown in Fig. 6 are simulations in which the same particle model and orientation are used across the atmospheric column. Therefore, particles are orientated the same regardless of shape, size, or altitude. This will likely reduce the variation in polarisation difference between 243 GHz and 664 GHz.

Therefore, it is difficult to conclude if, and how much, there will be a difference. In Kaur et al. (2022), simulations were performed at 166 GHz and 660 GHz. Although these are not the same frequencies as the ICI channels considered, the results are useful for comparison since a similar orientation scheme was used. Higher maximum polarisation differences were found at 166 GHz, despite using the same range of aARO factor and microphysics. We can perhaps draw a similar conclusion for ICI, i.e. higher polarisation differences could be found at 243 GHz when neglecting surface effects. However, we again stress that the setup of the simulations leads to less variation than might be present in reality. The above points have been added to the text, including the citation mentioned.

Line 623: A redundant bracket exists.
Fixed.

Line 646: the negative sign should be in $$.
Fixed.

Figure 13: The legend is incomplete; mid-latitude reference is missing
Fixed.

Line 799: EarthCare has been already launched; the authors could consider updating the information.
Fixed.

Line 801: remove "that"
Fixed.

Line 857: a redundant "-" exist
Fixed.

Line 858: a redundant "-" and a space exist.
Fixed.

**References**

Kaur, I., Eriksson, P., Barlakas, V., Pfreundschuh, S., and Fox, S.: Fast radiative transfer approximating ice hydrometeor orientation and its implication on IWP retrievals, Remote Sens., 14, 1594, https://doi.org/10.3390/rs14071594, 2022.